



# Three-channel single-wavelength lidar depolarization calibration

Emily M. McCullough[1,2a,*], Robert J. Sica[1], James R. Drummond[2], Graeme Nott[2,3a], Christopher Perro[2], and Thomas J. Duck[2]

[1]Department of Physics and Astronomy, The University of Western Ontario, 1151 Richmond St., London, ON, N6A 3K7
[2]Department of Physics and Atmospheric Science, Dalhousie University, 6310 Coburg Rd., PO Box 15000, Halifax, NS, B3H 4R2
[3]Facility for Airborne Atmospheric Measurements, Building 146, Cranfield University, Cranfield, MK43 0AL, UK
[a]Indicates present affiliation

*Correspondence to:* Emily McCullough (e.mccullough@dal.ca)

**Abstract.**

Linear depolarization measurement capabilities were added to the CANDAC Rayleigh-Mie-Raman lidar (CRL) at Eureka, Nunavut, in the Canadian High Arctic in 2010. This upgrade enables measurements of the phases (liquid versus ice) of cold and mixed-phase clouds throughout the year, including during polar night. Depolarization measurements were calibrated according

5    to existing methods using parallel- and perpendicular-polarized profiles as discussed in McCullough et al. (2017). We present a new technique that uses the polarization-independent Rayleigh elastic channel in combination with one of the new polarization-dependent channels, and show that for a lidar with low signal in one of the polarization-dependent channels, this method is superior to the traditional method. The optimal procedure for CRL is to determine the depolarization parameter using the traditional method at low resolution (from parallel and perpendicular signals), and then to use this value to calibrate the high-

10    resolution new measurements (from parallel and polarization-independent Rayleigh elastic signals). Due to its use of two high-signal-rate channels, the new method has lower statistical uncertainty, and thus gives depolarization parameter values at higher spatial-temporal resolution by up to a factor of 20 for CRL. This method is easily adaptable to other lidar systems which are considering adding depolarization capability to existing hardware.

## 1 Introduction

15    The Polar Environment Atmospheric Research Laboratory (PEARL, at 80 °N, 86 °W) has more than 25 instruments dedicated to in situ and remote sensing study of atmospheric phenomena in a location on Earth where few measurements are available. PEARL is located in Canada's High Arctic at Eureka, Nunavut. With climate changes magnified at such latitudes, PEARL's measurements give a valuable contribution to global atmospheric and environmental science.



The Candac Rayleigh-Mie-Raman Lidar (CRL) was installed in 2007 at PEARL (Nott et al., 2012). Linear 532 nm depolarization capabilities were added to the lidar in 2010 with the addition of a beam-splitter, a Licel Polarotor rotating polarizer, and a photomultiplier tube detector. McCullough et al. (2017) discusses the calibration and first results of this addition, using the depolarization parameter, $d$, found

using traditional methods. The depolarization parameter is the fraction of backscattered light which has become unpolarized through scattering interactions with the atmosphere (Gimmestad, 2008). Calculation methods in McCullough et al. (2017) were based on parallel-polarized (with respect to the outgoing laser plane of polarization as the beam exits the roof) and perpendicular-polarized measurement profiles which, at CRL, are made using a single PMT and a rotating prism which allows through light of each polarization

plane on alternate laser shots.

In the CRL, optics upstream of the depolarization channel act as a partial polarizer. The optics strongly attenuate the portion of the backscattered lidar intensity which is accepted by the perpendicular channel (which is half of any backscattered intensity which has become unpolarized), while attenuating by only a small amount the intensity which is accepted into the parallel channel (the other half of the backscattered

light which has become unpolarized, plus all backscattered intensity which remains polarized parallel to the transmission plane). The maximum signal in the parallel channel would be much greater than the maximum signal in the perpendicular channel, even without the partial-polarizer effects of the CRL's receiver optics. The CRL's optics exacerbate this effect by a factor of approximately 21 times (McCullough et al., 2017). This signal mismatch on the PMT, and very low signal rates in the perpendicular measurements,

are detrimental to traditional calculations of $d$. Traditional depolarization parameter calculations are simple to calibrate, but require long integration times and/or integration over large range scales (relative to the time and altitude scale of variation within the clouds) to produce acceptable uncertainties in the calculated values. The end result is an intermittent, relatively low resolution determination of $d$. The depolarization parameter determined in the traditional manner using the parallel and perpendicular measurements will,

in this paper, be called $d_1$.

The inclusion of an additional CRL measurement channel in the calculations proves helpful, and opens the possibility of a new calculation technique for determining $d$. Since 2007, CRL has included a polarization-independent Rayleigh elastic measurement channel at the same wavelength as the new depolarization channel. This polarization-independent channel has very high signal rates, and a high signal-to-noise ratio (SNR). It was posited that since all light backscattered to the lidar can be decomposed into parallel





and perpendicular components, that a linear combination of the signals in the parallel and perpendicular channels should be related to the signal measured in the polarization-independent Rayleigh elastic channel, which accepts light of all polarization planes. This would allow a measurement of $d$ which is not as dependent on the low SNR polarization-dependent measurements. The main advantages of the methods

presented here are as follows:

1. We can determine $d$ excluding the low-SNR polarization-dependent channel altogether.

2. We have the flexibility to include simultaneous information from the low-SNR polarization-dependent channel (the perpendicular channel for CRL) at low resolution to calibrate and improve the calculations of $d$ at high resolution from the high-SNR polarization-dependent channel (the parallel channel for CRL)

and the high-SNR polarization-independent channel.

We are not the first to propose a three-channel depolarization technique, but these other methodologies could not be implemented on the CRL measurements. Principally, this is because there is differential overlap between the CRL channels. Reichardt et al. (2003), henceforth R2003, uses the same three channels we propose here, but in characterizing the optical effects in each channel, accounts only for differences

in efficiency. They assume that all optical elements leading to each polarization analyzer have at most the action of a partial polarizer, and assumes that there is no differential overlap between any measurement channels. Their efficiency ratios $V_{1,2,3}$ (required to be "known" constants for the R2003 method) are, for CRL, functions of differential overlap, and therefore vary with altitude, lab temperature, and laser beam alignment. Freudenthaler (2016), henceforth F2016, describes detailed calibrations for a num-

ber of specific polarization lidar systems, some of which use a polarization-dependent channel with a polarization-independent channel, but none of which sufficiently describe the CRL system. Similar to the R2003 method, the methods in F2016 do not allow for the significant differential overlap contribution in the case of the CRL.

In both the R2003 method and the F2016 methods, all measurement channels are used simultaneously

at identical time and altitude resolutions, and no discussion is made of the impact of having one channel with much lower SNR than the others. The method shown in this paper allows for more flexibility in this regard, and can be adapted to many types of lidar systems.

Here, we present an extension to the Mueller Matrix algebra demonstrated in McCullough et al. (2017) for the parallel and perpendicular channels to the polarization-independent channel. We then show that it is

possible to determine the depolarization parameter $d$ using only the parallel and polarization-independent



channels, plus two calibration factors which must be measured. This scheme, which avoids use of the low signal-to-noise ratio perpendicular signals, yields a depolarization parameter with much higher spatial and temporal resolution than that produced by the traditional method. The disadvantage is that multiple calibration factors are required, at least one of which varies in altitude and time. When calculated using the new method, the depolarization parameter will be referred to as $d_2$.

Sky depolarization is neither dependent on the lidar nor on the way in which the lidar is calibrated, and thus $d_1 \equiv d_2$. We can therefore use the the intermittent low-resolution traditional depolarization parameter measurements ($d_1$) to determine the calibration factors required for the calculation of the depolarization parameter at high spatial and temporal resolution ($d_2$), including tracking the changes in space and time of the calibration factors. This scheme proves to be the most advantageous method for determining the depolarization parameter using the CRL lidar, or in general any lidar in which one of the polarized measurement channels has very low signal rates.

Examples of the calibration and calculation procedure for $d_2$, as informed by $d_1$, are provided for 10 March 2013, which highlight the advantage of the new method. A second example from 14 March 2013 shows some of the nuances in choosing a selection region for the $d_1$ values which are used in these calibrations based on atmospheric conditions. A more detailed examination of specific case studies using this method is available in McCullough's PhD thesis (2015).

The paper concludes with a discussion and suggestions for future work. The three-channel combined method advocated here is a powerful procedure which allows vastly improved depolarization parameter measurements at CRL, with lower uncertainty and higher spatial-temporal resolution, all with zero extra cost for equipment upgrades or negative impact on the other measurement channels in the lidar. The development shown here is easily adaptable to any similar lidar, and to any lidar with a single unpolarized, and single polarized channel.

## 2 Traditional depolarization method: Using Parallel and Perpendicular measurements to calculate $d_1$

Traditionally, the depolarization parameter $d$ is calculated using a combination of parallel and perpendicular polarized measurements, as in Eqn. (1) (e.g. Gimmestad (2008), and as used in McCullough et al.




(2017)):

$$d_1 = \frac{2kS_\perp}{S_\parallel + kS_\perp} = \frac{2k\frac{S_\perp}{S_\parallel}}{1 + k\frac{S_\perp}{S_\parallel}} = \frac{2}{\frac{1}{k}\frac{S_\parallel}{S_\perp} + 1}, \tag{1}$$

in which: $S_\perp$ is the corrected signal measured by the perpendicular channel, $S_\parallel$ is the corrected signal measured by the parallel channel, $k$ is the depolarization calibration constant, which is the ratio of the gains of the parallel and perpendicular channels. All signals $S$ have gone undergone the processes of cor-
5 rection for pulse pile-up (photon counting detection), correction for voltage scaling (analogue detection), merging of photon counting and analogue measurements into a combined profile, co-adding of profiles in time and altitude, and background subtraction. An example of such signals is shown in Fig. 1. In this work, the depolarization parameter calculated using this parallel-perpendicular method will be indicated as $d_1$. Fig. 2 provides some examples of $d_1$ as measured by CRL.

### 2.1 Calibration of $d_1$

Lamp and laser calibrations described in McCullough et al. (2017) introduce unpolarized light (simulating $d = 1$ from the sky) to the receiver. Solving Eqn. (1) for $k$, with $d_1$ set to unity, gives:

$$k = \frac{S_\parallel}{S_\perp}. \tag{2}$$

Measurements show that $k = 21.0 \pm 0.2$ for CRL. This value does not change from day to day. Indeed it has been shown to be stable at CRL for several years. It depends only on the partial polarizing effects of the
receiver's optical components and, in lidars which have separate PMTs for the parallel and perpendicular measurement channels, PMT gain.

### 2.2 Advantages and disadvantages to the traditional $d_1$ method

Calculations of $d_1$ are straightforward, requiring a single calibration factor which does not change from one measurement period to the next. However there are drawbacks to this method, specifically for CRL
lidar, and for any lidar for which the maximum parallel signal strength far exceeds the maximum perpendicular signal strength.

For CRL, low count rates in the perpendicular channel mean that much averaging in time and/or space is required to calculate $d_1$ to within an acceptable uncertainty. The user may decide which information (vertical spatial vs. temporal) is most important for addressing their scientific questions. Figure 1 gives



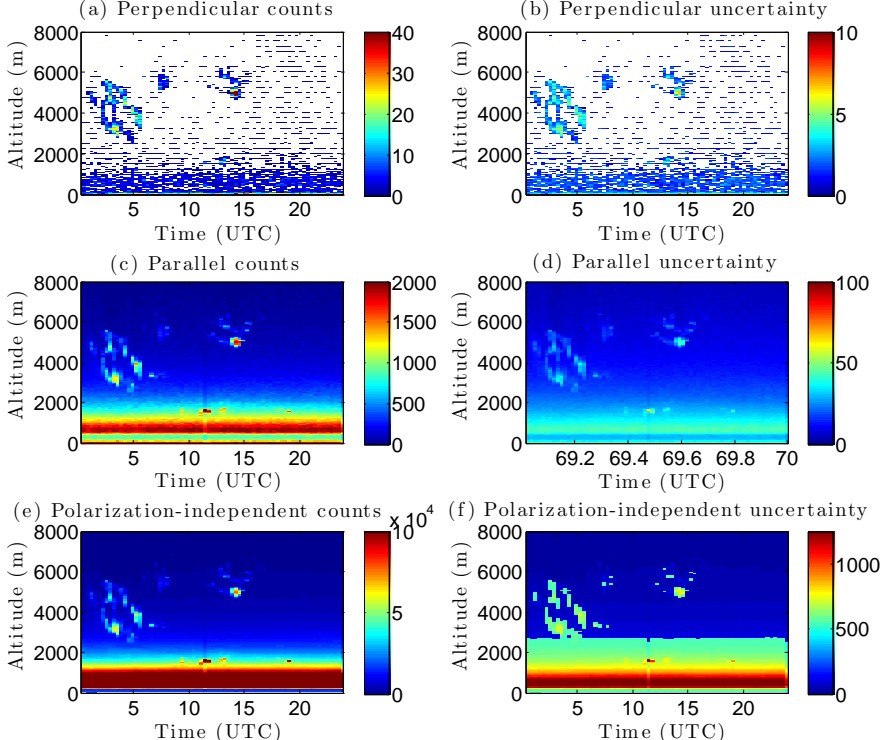

**Figure 1.** Corrected photocounts from 10 March 2013. The left column of panels gives the corrected photocounts for the perpendicular (top panel), parallel (centre panel), and polarization-independent Rayleigh elastic (lower panel) channels. The corresponding absolute uncertainties, in units of photocounts, are plotted in the right column of panels. All data points with a signal to noise ratio less than 1 have been removed and are coloured white. Because of the combined photon counting and analogue measurements at CRL, these uncertainties are not simply the standard deviation of the photocounts reported in the left column of plots. Resolution is $20\,\mathrm{min} \times 7.5\,\mathrm{m}$.

an example of coadded count rates in each channel at a resolution of $20\,\mathrm{min} \times 7.5\,\mathrm{m}$. At best, CRL can produce only values of $d_1$ with either low time resolution, or low altitude resolution. Cloud properties can change on the vertical scale of metres (e.g. for liquid layers within ice clouds), and minutes (depending on the speed of the clouds carried over the lidar's location), so the utility of cloud depolarization measurements is linked to the resolution at which they can be acquired. The general requirement for high spatial

and temporal resolution in determining cloud microphysical parameters such as liquid water content is stated by numerous authors. Requirements for sub-$100\,\mathrm{m}$ sampling are given by Mioche et al. (2017), Loewe et al. (2017), and Hogan et al. (2003), with requirements on the scale of $50\,\mathrm{m}$ given even earlier by Ramaswamy and Detwiler (1986) and Korolev et al. (2007), and recently by Sotiropoulou et al.





(2014) and Solomon et al. (2015). Averaging the depolarization parameter measurements over too large an area of time and space smears localised values of low and high depolarization to an appearance of a smooth region with an intermediate value. Incorrect interpretations of such over-averaged measurements are inevitable, as shown by the analysis of CALIOP satellite lidar measurements in Cesana et al. (2016).

Even with substantial co-adding of bins, there are frequently regions of the time-altitude measurement space (particularly at higher altitudes, where atmospheric density is lower) for which the CRL perpendicular measurement channel has too few counts to make a calculation. There are commonly measurement bins for which zero photons are measured, leading to intermittent calculated $d_1$ values.

## 3   Using Parallel and Polarization-Independent Rayleigh Elastic measurements to calculate $d_2$

The depolarization parameter may be calculated in an alternate manner using the parallel and polarization-independent Rayleigh elastic channels. In this work, the depolarization parameter calculated using the parallel and polarization-independent signals will be indicated as $d_2$. For CRL, count rates in each of these channels are much higher than the maximum count rates in the perpendicular channel, by a factor of 10 to 50 for parallel and by a factor of 200 for polarization-independent. Less co-adding leads to higher

resolution calculations of the depolarization parameter $d_2$.

McCullough et al. (2017) developed a full Mueller Matrix calculation for the signals in each depolarization channel. Under conditions which are true for the CRL lidar, these signal equations are expressed as Eqn. (3) and Eqn. (4):

$$S_{\parallel} = \frac{G_{PMT\parallel}bO_{\parallel\perp}(z)I_{laser}}{2}(M_{00} + M_{10} + (M_{01} + M_{11})(1-d)) \tag{3}$$

$$S_{\perp} = \frac{G_{PMT\perp}bO_{\parallel\perp}(z)I_{laser}}{2}(M_{00} - M_{10} + (M_{01} - M_{11})(1-d)), \tag{4}$$

in which: $S_{\parallel}$ and $S_{\perp}$ are the signal rates measured in the parallel and perpendicular channels, respectively, $G_{PMT\parallel}$ and $G_{PMT\perp}$ are the combined gains (or attenuations) of the focusing lens, interference filter, and photomultiplier tubes for each channel (where $G_{PMT\parallel} = G_{PMT\perp}$ for CRL as they share the same PMT

and associated optics), $b$ is an arbitrary gain factor used to normalize the atmospheric scattering matrix, $O_{\parallel\perp}(z)$ is the overlap function, containing all height-dependent variations in lidar signal, and named for the largest of these contributions which is the geometric overlap function, $I_{laser}$ is the laser intensity,


$M_{xx}$ are individual elements of the $4\times4$ Mueller matrix $\mathbf{M}$ which describes the combined optical effect of all optics in the shared beam path for the parallel and perpendicular channel which are upstream of the polarotor, and $d$ is the depolarization parameter of the atmosphere. The values $S_{\parallel}$, $S_{\perp}$, and $d$ are all understood to be functions of altitude, $z$, and time, $t$.

A similar argument to McCullough et al. (2017) may be made to develop and expression for the signal in the polarization-independent Rayleigh elastic channel. The matrix expression for the received Stokes vector, $I_R$, for the polarization-independent channel is given in Eqn. (5) and Eqn. (6), corresponding to McCullough et al. (2017) Eqn. (8). This channel does not contain a polarizer. Some optics such as the telescope and focus stage are in common for all channels, but others are different, for example, the visible long wave pass filter (VLWP): The polarization-independent Rayleigh channel receives a reflection off this optic while the parallel and perpendicular channels receive the transmission through it. Thus, the matrix for optics upstream of the polarization-independent PMT is given by $\mathbf{T}$, rather than $\mathbf{M}$ used for the polarized channels, $G_{PMTR}$ is used for the combined gains (or attenuations) of the focusing lens, interference filter, and photomultiplier tube associated with the polarization-independent channel, and $O_R(z)$ is used for the overlap function. $O_R(z)$ differs from $O_{\parallel\perp}(z)$ because of the different beam paths taken through the receiving optics of the instrument to reach each PMT, and the possibility that each channel focuses differently onto its PMT.

$$\boldsymbol{I}_R = G_{PMTR} \begin{pmatrix} T_{00} & T_{01} & T_{02} & T_{03} \\ T_{10} & T_{11} & T_{12} & T_{13} \\ T_{20} & T_{21} & T_{22} & T_{23} \\ T_{30} & T_{31} & T_{32} & T_{33} \end{pmatrix} bO_R(z) \begin{pmatrix} 1 & 0 & 0 & 0 \\ 0 & 1-d & 0 & 0 \\ 0 & 0 & d-1 & 0 \\ 0 & 0 & 0 & 2d-1 \end{pmatrix} I_{laser} \begin{pmatrix} 1 \\ 1 \\ 0 \\ 0 \end{pmatrix} \tag{5}$$

$$\boldsymbol{I}_R = G_{PMTR}bO_R(z)I_{laser} \begin{pmatrix} T_{00} + T_{01}(1-d) \\ T_{10} + T_{11}(1-d) \\ T_{20} + T_{21}(1-d) \\ T_{30} + T_{31}(1-d) \end{pmatrix}. \tag{6}$$

The signal rate $S_R$ in Eqn. (7) is the intensity element of the Stokes vector $\boldsymbol{I}_R$:

$$S_R = G_{PMTR}bI_{laser}(T_{00} + T_{01}(1-d)). \tag{7}$$

The goal is to determine an expression for the depolarization parameter using only the signals from the parallel and polarization-independent Rayleigh elastic channels, $S_{\parallel}$ and $S_R$. First, the polarization-



independent channel's signal equation (7) is solved for $bI_{laser}$ (a quantity which can not be truly known during any given measurement and thus we desire to eliminate it):

$$bI_{laser} = \frac{S_R}{G_{PMTR}O_R(z)} \frac{1}{T_{00} + T_{01}(1-d)}. \tag{8}$$

Substituting Eqn. 8 into the parallel channel's signal equation (3) and solving for the depolarization parameter (now labelled $d_2$):

$$S_\| = \frac{G_{PMT\|}O_{\|\perp}(z)}{2} \frac{S_R}{G_{PMTR}O_R(z)} \frac{M_{00} + M_{10} + (M_{01} + M_{11})(1-d_2)}{T_{00} + T_{01}(1-d_2)} \tag{9}$$

$$d_2 = 1 + \frac{\frac{1}{2}\frac{G_{PMT\|}O_{\|\perp}(z)}{G_{PMTR}O_R(z)}\frac{S_R}{S_\|}(M_{00} + M_{10}) - T_{00}}{\frac{1}{2}\frac{G_{PMT\|}O_{\|\perp}(z)}{G_{PMTR}O_R(z)}\frac{S_R}{S_\|}(M_{01} + M_{11}) - T_{01}} \tag{10}$$

$$d_2 = 1 + \frac{\frac{1}{2}\frac{S_R}{S_\|}\left(1 + \frac{M_{10}}{M_{00}}\right) - \left(\frac{G_{PMTR}O_R(z)}{G_{PMT\|}O_{\|\perp}(z)}\frac{T_{00}}{M_{00}}\right)}{\frac{1}{2}\frac{S_R}{S_\|}\left(\frac{M_{01}}{M_{00}} + \frac{M_{11}}{M_{00}}\right) - \left(\frac{G_{PMTR}O_R(z)}{G_{PMT\|}O_R(z)}\frac{T_{01}}{M_{00}}\right)}, \tag{11}$$

in which: $S_\|$ and $S_R$ are measurements, while $M_{xx}$, $T_{xx}$, $G_{PMT\|}$ and $G_{PMTR}$ must be determined by calibration measurements. Overlap functions are in general difficult to determine for lidars. Here, the "overlap function" $O(z)$ includes both geometric overlap (varies in altitude and time) as well as any other factors which vary in altitude (though they may be constant in time). The overlap function will be

eliminated where possible, and available means will be used to determine it via calibration otherwise.

Five calibration factors are thus needed: $\frac{M_{01}}{M_{00}}$; $\frac{M_{10}}{M_{00}}$; $\frac{M_{11}}{M_{00}}$; $\frac{G_{PMTR}O_R(z)}{G_{PMT\|}O_{\|\perp}(z)}\frac{T_{00}}{M_{00}}$; and $\frac{G_{PMTR}O_R(z)}{G_{PMT\|}O_{\|\perp}(z)}\frac{T_{01}}{M_{00}}$. Some information is already known: from polarized and unpolarized white light characterization tests in McCullough et al. (2017), which found $\frac{M_{01}}{M_{00}} = 0.91 \pm 0.002$ for CRL. Thus, each channel has a different gain, indicated by $M_{01} \neq M_{00}$. Further, $M_{11} = M_{00}$ and $M_{01} = M_{10}$, indicating an absence of cross-talk between

the parallel and perpendicular channels; no parallel-polarized light gets into the perpendicular profiles, and vice versa. Detailed characterizations carried out with polarized light introduced to the receiver at a variety of angles show that if there is any sensitivity to polarization in the "polarization-independent" Rayleigh elastic channel, this effect is orders of magnitude smaller than the uncertainty in routine lidar measurements and does not affect analyses (Appendix A). As the CRL polarization-independent Rayleigh

elastic channel has been shown to be insensitive to changes in polarization (i.e. responds independently of polarization of incoming light), $T_{01} = 0$. Were this not the case, its signal would depend on the depo-





larization effects of the atmosphere. Therefore, the equation for $d_2$ simplifies to:

$$d_2 = 2 - \left( \frac{2}{1 + \frac{M_{10}}{M_{00}}} \right) \left( \frac{G_{PMT R} O_R(z)}{G_{PMT \parallel} O_{\parallel \perp}(z)} \frac{T_{00}}{M_{00}} \right) \left( \frac{S_\parallel}{S_R} \right). \tag{12}$$

None of $M_{00}$, $M_{10}$, and $T_{00}$ is needed individually. Nor is any individual overlap function $O(z)$ required, although a ratio of these is included. The ratio of overlap functions is unlikely to be stable in time, and this must be taken into account when calibrating. We require only two calibration factors: $\frac{M_{10}}{M_{00}}$, which is stable in time and has already been determined, and $\frac{G_{PMT R} O_R(z)}{G_{PMT \parallel \perp} O_{\parallel \perp}(z)} \frac{T_{00}}{M_{00}}$, which can vary and will likely require more frequent calibrations. For clarity, we define a new variable $Y(z) = \frac{G_{PMT R} O_R(z)}{G_{PMT \parallel \perp} O_{\parallel \perp}(z)} \frac{T_{00}}{M_{00}}$, such that

$$d_2 = 2 - \left( \frac{2}{1 + \frac{M_{10}}{M_{00}}} \right) Y(z) \left( \frac{S_\parallel}{S_R} \right). \tag{13}$$

## 3.1 Calibration of $d_2$

The calibration profiles $Y(z)$ must be determined before $d_2$ can be calculated. Unlike all other calibration terms in the equation for depolarization parameter using the $d_2$ setup, $Y(z)$ may vary with altitude. It contains the overlap functions $O_{\parallel \perp}(z)$ and $O_R(z)$ in a ratio indicating the differential overlap between the polarization-independent and the depolarization photomultiplier tube viewing geometries. Equation (13) is solved for the calibration profile:

$$Y(z) = \frac{1}{2} \left( 1 + \frac{M_{10}}{M_{00}} \right) \left( \frac{S_R}{S_\parallel} \right) (2 - d_2) \tag{14}$$

In order to set $d_2 = 1$ in Eqn. (14), enabling us to solve for $Y(z)$, a glassine waxed paper depolarizing sheet is placed over the lidar's roof window, which depolarizes all backscattered light as it enters the lidar receiver (McCullough et al., 2017). The lidar is then operated as normal, using the laser beam backscattered from the sky as a light source. A lamp will not suffice for this calibration, because of the altitude-dependence of the calibration profile we seek. The calibration calculation with the depolarized-lidar setup becomes:

$$Y(z) = \frac{1}{2} \left( 1 + \frac{M_{10}}{M_{00}} \right) \left( \frac{S_R}{S_\parallel} \right). \tag{15}$$

Since it is possible to change gain settings on the parallel PMT from time to time, it may be desirable to keep $\frac{M_{01}}{M_{00}}$ as a term in $d_2$ calibrations. In that case, an identical unpolarized white light calibration may



be done while solving Eqn. (13) for the entire term $\left( \frac{2Y(z)}{1+\frac{M_{10}}{M_{00}}} \right)$. This large calibration term could then be applied to measurements taken the same day as the calibration. This possibility is not explored further here, as $\frac{M_{01}}{M_{00}} = 0.91 \pm 0.002$ was not changing for CRL during the measurement period in question.

## 3.2 Advantages and disadvantages of $d_2$

There are more practical considerations for the $d_2$ calibration than there are for the $d_1$ calibration. The glassine sheet attenuates all signals, and it is important to have calibration measurements from all relevant lidar heights (for CRL preferably up to 10 to 20 km altitude). Thus, particular atmospheric situations are helpful during the calibration, especially those with highly backscattering clouds at mid and high altitudes. It takes several hours to do this measurement to build up a good calibration profile.

A critical disadvantage is that the "constant" profile $Y(z)$ contains overlap functions which could change with time, such as each time the laser beam is realigned to the sky, and when there is a change in laboratory temperature. Therefore $Y(z)$ must be determined *each night* (unless experience shows that a less frequent calibration suffices), by putting a depolarizing sheet over the lidar, accumulating sufficient counts (which are attenuated during the calibration) to determine the calibration profile, then removing the sheet, and making actual measurements of the atmosphere for the remainder of the night. Realistically, CRL can be calibrated in this way or it can measure the atmosphere, but not both in any given night. If the calibration profiles are determined to be *sufficiently constant* from night to night, this calibration method could be used every couple of days in between days of good measurements. (In this case, an uncertainty will be introduced to $d_2$, which can be estimated by examining the typical variation in profiles of $Y(z)$ and propagating this value through the equations for $d_2$. Each lidar retrieval's tolerance for additional uncertainty in its $d_2$ calculations will determine the level of variation which can be tolerated in $Y(z)$).

While possible for a lidar with a local operator, this procedure is not practical for a remotely operated instrument such as CRL. Nonetheless, $d_2$ offers attractive advantages due to the higher signal rates involved: The resolution of $d_2$ far exceeds that of $d_1$ for CRL, and measurements are available to higher altitudes. It is possible circumvent around $d_2$'s calibration disadvantages by informing the calculation using $d_1$ values from the same measurement period. This procedure is discussed in the following section, recalling that both $d_1$ and $d_2$ represent the true depolarization state of the atmosphere, $d$.





## 4 Combining methods: Using low resolution $d_1$ to initialize high resolution $d_2$

Calibration constants required for calculations of $d_1$ and of $d_2$ may all be determined through special cal-
ibration measurements with the lidar (McCullough et al. (2017) for $d_1$, and Sect. 3.1 for $d_2$, respectively).
This provides two nearly independent results for the depolarization parameter for a particular measure-
ment period: $d_1$ with a well-understood calibration constant, but with low resolution values, and $d_2$ with
more complicated calibration constants which can change over time, but with higher resolution values and
more coverage in space and time.

A more advantageous approach is to combine the efforts of these two methods, using the low-resolution
$d_1$ values to inform the high resolution $d_2$ values for that same day's measurement. In effect, high resolu-
tion $d_2$ values can be calibrated daily using low resolution $d_1$ values rather than using a separate special
calibration procedure.

### 4.1 Calibration method using $d_1$ to determine $Y(z)$ and apply it to measurements of $d_2$

It is possible to solve initially for the depolarization parameter $d_1$ at high altitude resolution, but low
time resolution, feed it into the calibration Eqn. (14), and solve for a nightly calibration profile. Then this
profile can be used in the expression for $d_2$ (Eqn. (13)), even at resolutions not possible with the original
$d_1$ measurements. Calibrations to determine the $Y(z)$ profile may be carried out during any reasonable
period which contains simultaneous measurements from all three channels: parallel, perpendicular, and
polarization-independent. The methodology is as follows:

1. Parallel and perpendicular measurements are used to determine the depolarization parameter $d_1$ using
   the traditional method at high altitude resolution, but low time resolution. Many data points may still
   be missing, because perpendicular count rates are low.

2. Eqn. (14) is used to calculate $Y(z)$ at the same high altitude resolution, but low time resolution, as
   the calculations made for $d_1$. In contrast to the method in Sect. 3.1, this time no special hardware
   is put in place, and therefore $d_2$ is not set to unity in the equation (i.e. we cannot use Eqn. (15).
   The depolarization parameter of the sky at any space and time remains physically the same quantity
   whether it is measured via the traditional method ($d_1$) or via the new method ($d_2$). Thus, the $d_1$ values
   themselves are fed into Eqn. (14) as the values for $d_2$ in order to determine the $Y(z)$ values at each
   time and altitude.





3. These $Y(z)$ values are combined to create a single calibration profile of $Y(z)$ for the measurement period at high altitude resolution.

4. Parallel and polarization-independent measurements are then coadded to their optimal time and altitude resolutions. Their higher photocount rates mean that less coadding is required to achieve the same signal to noise that is possible with the perpendicular measurements at low resolution.

5. Finally, the single $Y(z)$ profile for the night is applied to each profile of the high resolution parallel and polarization-independent signals to calculate $d_2$. The result is that the $d_2$ values calculated in this way can have higher resolution, and retain more data points, than $d_1$.

The method described here is most advantageous for CRL's depolarization calculations, as demonstrated in the following sections.

## 5   Measurements to demonstrate the three-channel method

A night of regular-operations measurements (i.e. not a special calibration run) on 10 March 2013, with measurements made in all three channels (parallel, perpendicular, and polarization-independent), is used for this demonstration.

### 5.1   Signals and uncertainties in each channel

The night of 10 March 2013 was clear below $3500\,\mathrm{m}$ with, several clouds above this height. The clouds are not particularly thick; signal is visible above each of them in the parallel and polarization-independent Rayleigh elastic channels. The entire night's measurements and associated uncertainties are shown in Fig. 1. The plots here have been photon counting dead-time corrected, analogue range scaled and dark count corrected, and have been coadded and background subtracted. Coadding resolution was chosen to be 20 time bins ($20\,\mathrm{min}$) and 1 altitude bin ($7.5\,\mathrm{m}$) in order to have sufficient perpendicular signal for analyses while retaining as much vertical resolution as possible. Photocount rates in the perpendicular channel are exceeded by those in the parallel channel by a factor of between 10 and 50 times, and by those in the polarization-independent Rayleigh elastic channel by a factor of approximately 200. Consequently, the signal-to-noise ratios in the latter two channels are far superior to that in the perpendicular channel. The absolute uncertainties include the statistical measurement uncertainties carried through the described





processing using standard error propagation methods. Because of the combined photon counting and

analogue measurements at CRL, these uncertainties are not the simply the standard deviation of the photocounts reported in the top row of plots, although this element is the dominant contributor to the overall uncertainty values. The statistical uncertainty of the merged profiles has been discussed in McCullough (2015).

### 5.2   Depolarization parameter $d_1$ as determined by the traditional method

The depolarization parameter $d_1$ is determined using the parallel and perpendicular measurements following Eqn. (1), resulting in the upper panel of Fig. 2. The associated absolute uncertainty is calculated using standard error propagation equations, and is shown in the centre panel of Fig. 2. For example, the measurements at 03:30:00 UTC at 3000 m altitude have a depolarization ratio of approximately $d_1 = 0.6 \pm 0.1$ as read from the upper and centre panels. To calculate the relative uncertainty shown in the lower panel of

Fig. 2, the absolute uncertainty was divided by the associated value of $d_1$ and expressed as a percent; thus, the same measurement as read from the upper and lower panels is approximately $d_1 = 0.6 \pm 17\,\%$. Both expressions for uncertainty are useful in interpreting the depolarization values. The high depolarization parameter values at 04:00:00 UTC, at 5500 m altitude, indicate that the cloud is composed of particles which are not homogeneous spheres; in context, this means that the cloud is likely composed of ice par-

ticles. The uncertainty in $d_1$ in this region of the cloud is approximately 12 %. For the small cloud at 07:30:00 UTC, there is less certainty. There, the $d_1$ values indicate a mix of high and low depolarization varying between 0.4 and 0.8 in a rather noisy fashion. The uncertainty in this small cloud is $\pm 0.25$ or higher, indicating more than 30 % relative uncertainty. The edges of all clouds have high uncertainty as well. While a general interpretation of icy clouds in a clear atmosphere is possible, depolarization parameter measurements with higher resolution and/or smaller uncertainty would be better. If there are clouds above 6000 m altitude, $d_1$ is not sensitive to them because of the low count rates in the perpendicular channel. The extremely low signal rates in the perpendicular channel lead to many time-altitude points having insufficient signal-to-noise ratios to be considered (S/N < 1). Consequently, much of the time-altitude

5   space in the plot of $d_1$ is blank.




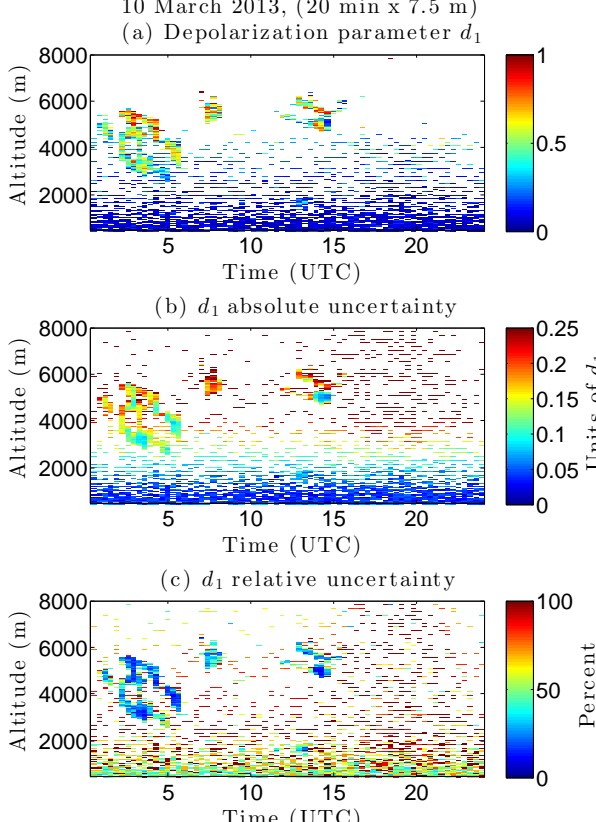

**Figure 2.** Top panel: Traditional method $d_1$ depolarization values from 10 March 2013. Centre panel: Absolute uncertainties associated with the $d_1$ values, in units of depolarization parameter. Lower panel: Relative uncertainties, in units of percent. Any locations with photocount signal to noise ratios smaller than 1 have been removed and are coloured white. No points have been removed based on calculated uncertainty in $d_1$.

## 5.3 Determining the calibration profile $Y(z)$

Next, the polarization-independent channel is brought into the evaluation, and the calibration profile $Y(z)$ is determined.





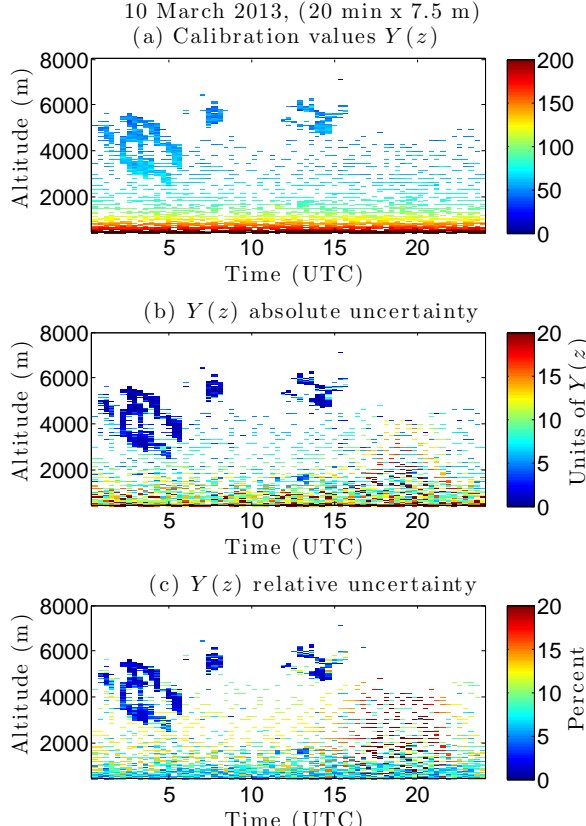

**Figure 3.** Top panel: The $Y(z)$ calibration values from 10 March 2013 individually for each data point. Centre and lower panels: absolute and relative uncertainties, respectively, in the calculated individual values of $Y(z)$.

### 5.3.1 Calculations of $Y(z)$ for each data point

Using Eqn. (14), a value of $Y(z)$ is determined for each data point in time and altitude, based on the $d_1$ values shown in Fig. 2. The results and their uncertainties are given in Fig. 3. The uncertainties are calculated assuming uncorrelated errors, using standard error propagation methods.

If there was good signal in the perpendicular channel, and thus good calibration measurements for each altitude, it would be possible to calibrate the lidar measurements scan-by-scan. However, there are frequently too few perpendicular measurements to make good statistics in this manner. It is feasible to determine one single profile for the night (perhaps even persisting longer) to use as a function of $Y(z)$ with altitude. The differential overlap function may be changing with such factors as temperature of the





lab, laser alignment, etc. Differential overlap is commonly known to vary little within one night, varying more between nights, particularly if the lidar has been cooled down and warmed back up in the interim, or if the lidar has been realigned to the sky. The latter procedure is carried out at the beginning of each very clear night. Conversely, the $\frac{G_{PMTR}}{G_{PMT\|\perp}}$ and $\frac{T_{00}}{M_{00}}$ portions of $Y(z)$ should be relatively stable provided no instrument parameters are changed. Therefore, the combined calibration factor $Y(z)$ is expected to vary slowly with time following the temporal trends of the differential overlap function.

### 5.3.2 Combining individual measurements into a single $Y(z)$ for the night

First, a mean profile in altitude is taken based on the calculated individual values. The propagated uncertainty reduces drastically as a large number of profiles are combined (just as for any co-adding procedure). A smooth profile was desired so that the profile would not be unduly influenced by small clouds, etc. A 10-point moving-average filter was applied to the mean profile to smooth it in altitude.

A number options were tested to determine the optimal profile of $Y(z)$ with altitude, and acceptable results were found using a powerlaw fit to the entire profile, as shown in Fig. 4. A power law of the form $y = ax^b + c$ was found to fit the calibration data ($y$, the smoothed mean profile) with altitude ($x$) with goodness-of-fit $R^2$ of greater than 0.998 in every case studied. For the example data shown in this section, the coefficients to the power law are given by $a$, $b$, and $c$, with 95 % confidence bounds in brackets:

$a = 1.152 \times 10^5 (1.082 \times 10^5, 1.223 \times 10^5)$

$b = -1.026(-1.036, -1.017)$

$c = 31.81(31.29, 32.34).$

This fit has $R^2 = 0.998$ and the root mean square error is RMSE $= 1.523$ (compared to the values of $Y(z) = 400$ at its largest point, and around 40 to 50 at its smallest).

In Fig. 4, four curves are plotted over the individual profiles: the mean profile; the upper and lower bounds on the mean profile based on the mean profile's uncertainty; the power-law fit function; and the upper and lower bounds on the power law fit function based on the root mean square error in the fit itself. Note that the root mean square error in the fit dominates the error in the mean profile. It was determined that the measurement error in the mean profile could be neglected in the fitting process for this reason. This quantity is quite stable over the course of the night, indicated by the near coincidence of all profiles plotted in each panel of Fig. 4, and the lack of a trend in time at any altitude in Fig. 3.





$Y(z)$ Calibration profiles, 10 March 2013 (20 min x 7.5 m)

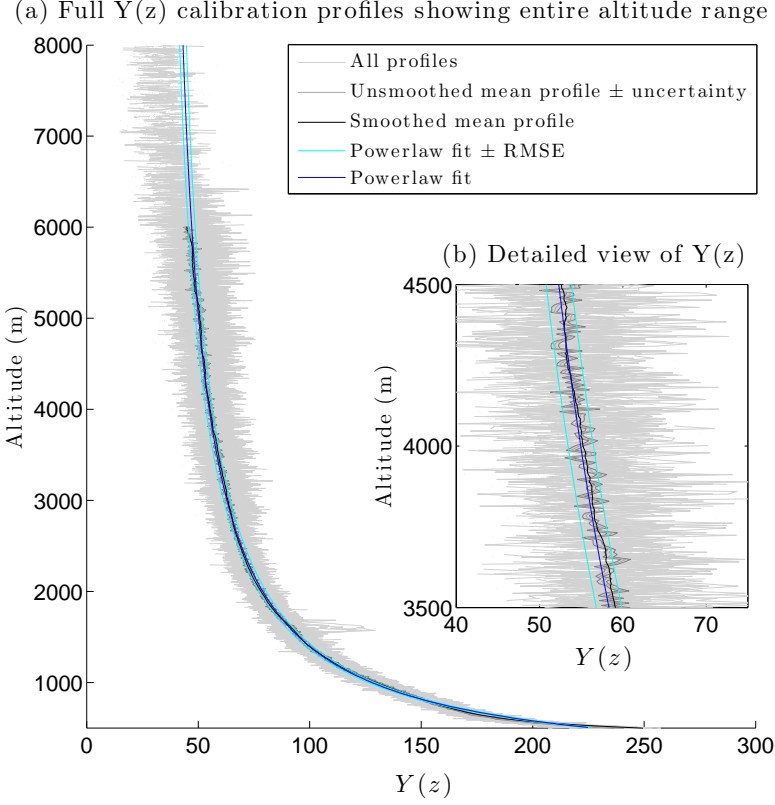

**Figure 4.** Panel a: The $Y(z)$ calibration values from 10 March 2013 with the various fits to the night's measurements. Panel b, inset: Zoomed-in portion of the plot on the left to show differences between the lines.

The residuals for the powerlaw fit for 10 March 2013 are given in Fig. 5. For the altitudes used in CRL analysis, there is no trend in the residuals, and therefore the powerlaw fit is acceptable. In the left panel, the residuals are given as the difference between the individual $Y(z)$ profiles and the nightly powerlaw fit. Each of these residual profiles can also be expressed as a percent residual profile (not shown). Below 3500 m altitude, the residuals are always smaller than the mean percent uncertainty in the individual $Y(z)$ profiles themselves, and the values are comparable up to 4500 m. This indicates that the powerlaw fit is at least as good, below these altitudes, as any of the individual profile measurements themselves. The large amounts of scatter and bias above 6000 m can be explained by the paucity of valid data points at those





5 altitudes. The right panel of Fig. 5 indicates fewer than 5 valid points going into the $Y(z)$ calculations at each of those altitudes. This is the region above the topmost clouds visible in Fig. 3. Everywhere below 6000 m, the residuals are smaller than 20 % of the value of the nightly powerlaw $Y(z)$ profile.

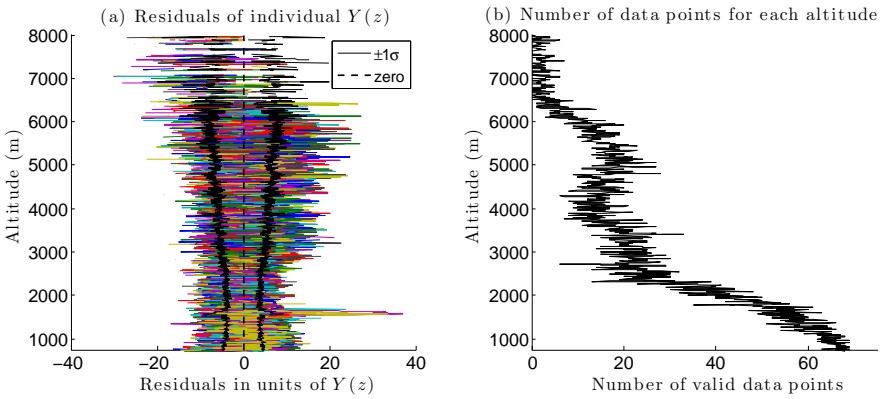

**Figure 5.** Left panel: The residuals of individual $Y(z)$ profiles for 10 March 2013 are shown after subtracting the nightly powerlaw fit. Solid black lines indicate $\pm 1\sigma$ of the measured profiles. The dashed line at zero emphasizes the lack of bias in the fit. Right panel: The number of valid data points for $Y(z)$ at each altitude. More data points are available anywhere photon signals are high, such as at low altitudes and inside clouds. The deviation of the residuals from zero in the left figure above 6000 m is explained by the lack of data points above this altitude.

### 5.3.3 Variation in the profile $Y(z)$ with changing co-adding resolution and with different dates

10 To check whether the profile of $Y(z)$ with altitude is different depending on the co-adding of the original data, the calibration procedure was carried out for the following resolutions of 10 March 2013 data: $(10 \min \times 7.5 \, \text{m})$, $(20 \min \times 7.5 \, \text{m})$, and $(10 \min \times 37.5 \, \text{m})$. To check whether the profile changes with time on scales longer than one day, data from 11 March 2013 and 14 March 2013 were also examined, mostly at $20 \min \times 7.5 \, \text{m}$ resolution. The general form of these fits is unchanging for these days in March 2013, as shown in Fig. 6. This suggests that it is appropriate to use the calibration profile from one day to make $d_2$ measurements from a nearby day. This could be useful if the perpendicular channel

5 is unavailable for one day for some reason. Also, there are certain sky conditions which are not well-suited for the determination of the calibration profile. Sect. 6 contains an example of one measurement





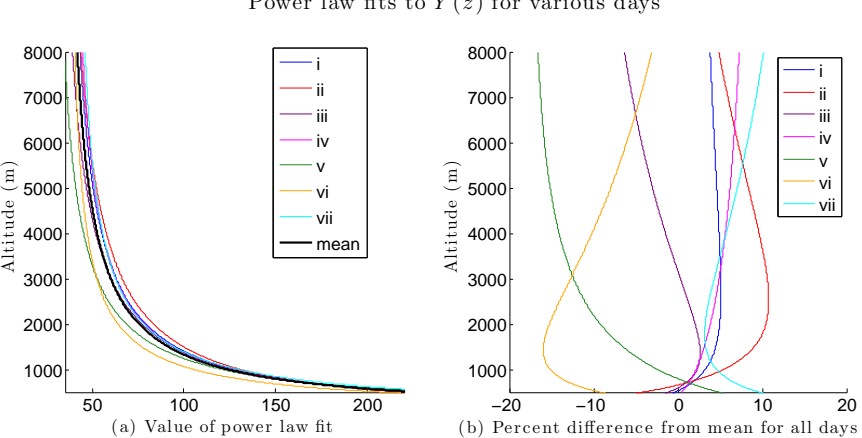

**Figure 6.** Left: Plots of 2nd order power law fits to the calculated profiles of $Y(z)$ for the test dates and resolutions listed in Table 1. Right: Residuals of each profile from the mean powerlaw fit profile. Compare this panel to the centre panel of Figure 5; the residuals are of similar magnitude to the uncertainties in the individually measured $Y(z)$ profiles.

day containing just such a situation. In that case, a nearby day's calibration may be preferable to its own day's calibration.

### 5.4 Determinations of $d_2$ at low resolution

A sample from 10 March 2013 is given in Fig. 7, with $d_2$ calculated using Eqn. (13) and the calibration profiles discussed in Sect. 5.3.2. The resolution is kept at $20\,\mathrm{min} \times 7.5\,\mathrm{m}$, the same as it is for the calculation of $d_1$.

The values retrieved for $d_2$ as informed by $d_1$ give results $0 \leqslant d_2 \leqslant 1$ as required. Regions of high depolarization parameter are visible within the clouds, as they are for the traditional $d_1$ results in previous plots. These regions also have low absolute uncertainty, as do the very low altitudes where the density of the atmosphere is large. These plots of $d_2$ also show much better coverage of the space and time region in question; fewer data points are missing to low signal-to-noise. There is now meaningful depolarization parameter information in all regions of the plot.

While data for $d_1$ ends entirely at around $2000\,\mathrm{m}$ altitude, except in the cloud, data for $d_2$ extends to above $5000\,\mathrm{m}$ with uncertainty smaller than $\pm 0.25$, and less certain values are calculable at yet higher altitudes. The cloud features are better delineated with $d_2$ data, and have lower uncertainty than those calculated as $d_1$.





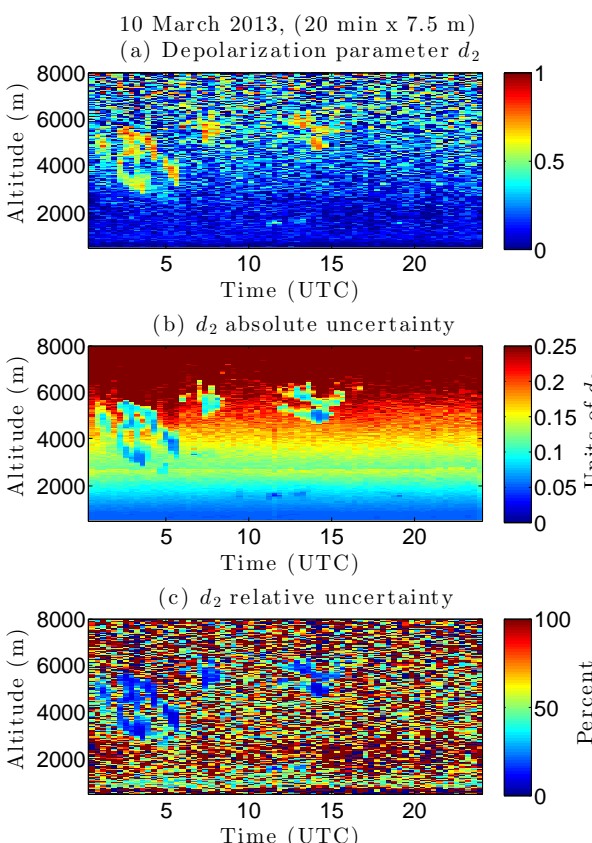

**Figure 7.** Top panel: The $d_2$ depolarization values from 10 March 2013, calculated at the same resolution ($20\,\text{min} \times 7.5\,\text{m}$) as the $d_1$ values shown in Fig. 2. Higher resolution $d_2$ results follow in Fig. 10. Centre panel: The uncertainties associated with the $d_2$ values are all in units of depolarization parameter. Lower panel: The relative uncertainties are in units of percent. The current figure demonstrates two advantages of the $d_2$ method over the $d_1$ method when the resolution is kept constant: better coverage of time and space, and lower uncertainty for each data point.



Some regions do have $d_2$ relative uncertainty approaching 100 %, but these are regions with depolarization parameter values < 0.1. A measurement of $d = 0.1 \pm 0.1$ is still a meaningful value indicating non-ice particles, or clear air. The values of $d_1$ in the clouds at 04:00:00 UTC, 5000 m have, in some locations, values of $d = 0.6 \pm 0.25$. This is a less definitive measurement in terms of interpretation since the low end of the uncertainty range would indicate liquid particles while the high end of the uncertainty range would indicate ice. On the other hand, the same cloud regions in the $d_2$ plot have values of $d = 0.6 \pm 0.07$ and a relative uncertainty less than 10 %, which is a clear indication of ice.

The one location that $d_1$ appears better constrained than $d_2$ is the horizontal strip along 1000 m altitude. Examining the uncertainty plots, $d_1$ has less than 0.04 absolute uncertainty in this location, while $d_2$'s is closer to 0.06. The reason for this is the use of the analogue counting channel in the parallel and polarization-independent counts which go into creating $d_2$. As the processing routine switches over from photon counting to analogue detection, the larger analogue uncertainty becomes visible. Further refinement of the processing routines may help in this regard.

## 5.5   Comparing the two methods: $d_2$ reproduces $d_1$

To ensure that the results for $d$ calculated using the new method ($d_2$; Fig. 7) are valid, they must be compared with those calculated using the traditional method ($d_1$; Fig. 2) and have the same values to within the uncertainty of both of the measurements.

To make the comparison, each plot had the following steps applied in sequence: 1. Removal of any data points for $d_1$ and $d_2$ with absolute uncertainty greater than 0.2; 2. Smoothing by $3 \times 3$ moving average filter, for a smoothed resolution of $60\,\mathrm{min} \times 22.5\,\mathrm{m}$; 3. Removal of any data points which are surrounded on three or four sides by an empty data point, done recursively twice such that isolated groups of two data points will also be eliminated; 4. Removal of any data points which do not exist in both of the plots.

A scatter plot can then be created of all $(d_1, d_2)$ pairs. For clarity, this information is represented by plotting the natural logarithm of the number of data points present at each $(d_1, d_2)$ location, binned in a 0.02 x 0.02 grid. In this way, we see the overall trend of the measurements. This is shown in Fig. 8. Red points indicate that more than sixty $(d_1, d_2)$ pairs lie at that location; Blue points indicate the locations of fewer than two $(d_1, d_2)$ pairs. The black line in this figure is a 1:1 line for reference. It is not a regression line, but does demonstrate the 1:1 trend between $d_1$ and $d_2$ values.





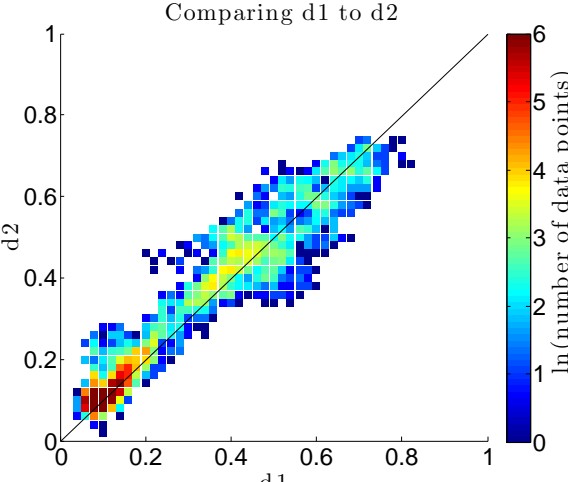

**Figure 8.** Scatter plot of $d_1$ versus $d_2$, binned with a grid of 0.02 x 0.02. Colour of point gives ln(number of data points at that particular $(d_1,d_2)$ grid location). A 1:1 line is also given. This is not a regression, but demonstrates the basic equivalence of the two methods.

In most altitude and time bins, there is no difference between the depolarization parameters calculated to within the limits of their combined uncertainties. Using data from the original $d_1$ and $d_2$ plots (Fig. 2 and Fig. 7, after doing only steps 1 and 4: removing points with uncertainty $> 0.2$, and removing any points which do not exist in both $d_1$ and $d_2$, but doing no smoothing nor removing of isolated data points), 14705 of 16024 valid data points match to within their uncertainties (91.8 %). Using data which has been processed through all four steps, including smoothing and removing isolated data points (such as the data used to make Fig. 8, this improves to 12941 of 13036 points (99.3 %). This is encouraging, as it shows that the CRL's parallel and polarization-independent method ($d_2$) is as valid as its parallel and perpendicular method ($d_1$).

These tests indicate that $d_1$ and $d_2$ are similar for almost all the times when they can be measured simultaneously, giving confidence that $d_1$ and $d_2$ are the same quantity and that $d_2$ values with their increased spatial and temporal coverage can be relied on.

In almost all situations, this $d_2$ procedure provides measurements of $d$ with significantly reduced uncertainty as compared to the $d_1$ procedure which relies on the perpendicular channel. The following sections demonstrate the true power of the $d_2$ method: access to depolarization ratio measurements at much higher resolution than is possible with $d_1$.



### 5.6 Determinations of $d$ at higher resolution

Using the traditional method, $d_1$ is calculated a higher resolution of $10\,\mathrm{min} \times 7.5\,\mathrm{m}$, keeping only data for which photon count signal-to-noise > 1 and for which absolute $d_1$ uncertainty < 0.2. The resulting plots in Fig. 9 readily show a deterioration in interpretability as compared to the plot using the lower resolution $20\,\mathrm{min} \times 7.5\,\mathrm{m}$ values of $d_1$ (calculated in Sect. 5.2). There are large differences in data coverage at this higher resolution, and there are atmospheric features which are no longer able to be discerned.

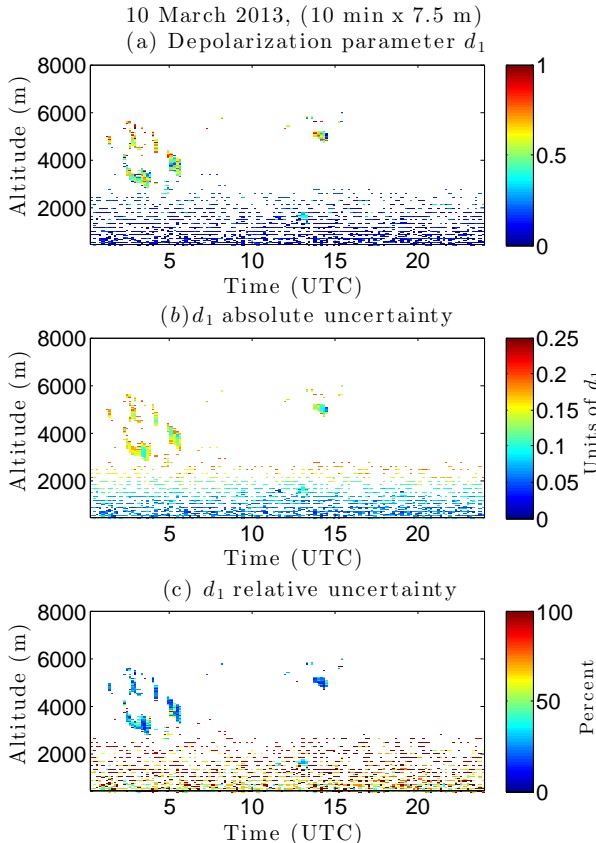

**Figure 9.** The $d_1$ depolarization values from 10 March 2013, with $10\,\mathrm{min} \times 7.5\,\mathrm{m}$ resolution, excluding anywhere with more than 0.2 absolute uncertainty in units of depolarization parameter. Top panel: $d_1$ values. Centre panel: The uncertainties associated with the $d_1$ values in units of depolarization parameter. Lower panel: The relative uncertainties are in units of percent.



Conversely, a calculation of $d_2$ at the higher resolution remains meaningful. The example illustrated here uses the low resolution $20\,\text{min} \times 7.5\,\text{m}$ values of $d_1$ (calculated in Sect. 5.2) and the resulting $Y(z)$ calibration profile (calculated in Sect. 5.3.2). These low resolution calculated values are then applied to parallel and polarization-independent photocount measurements at twice the time resolution: $10\,\text{min} \times 7.5\,\text{m}$. The resulting higher resolution values of $d_2$ are given in Fig. 10, retaining in the plot all data for which absolute uncertainty is less than 0.2.

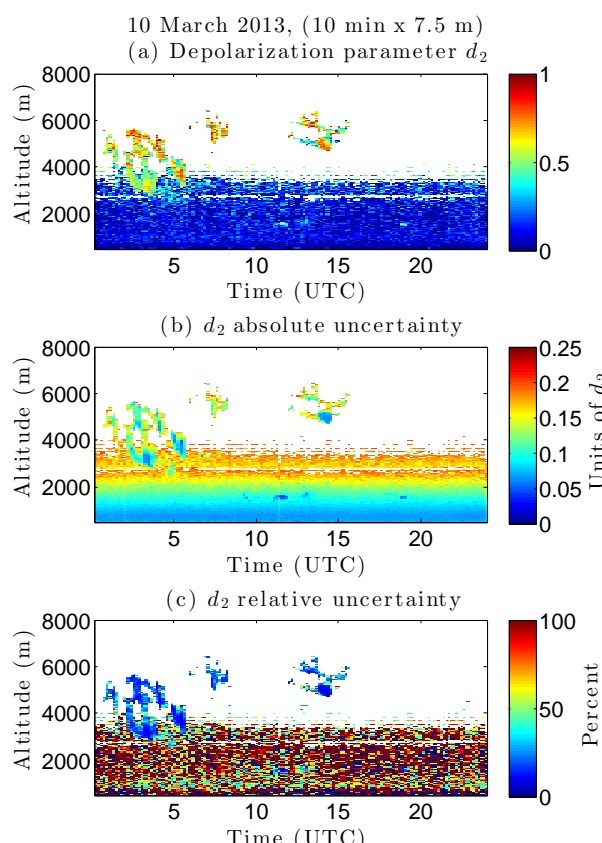

**Figure 10.** The $d_2$ depolarization values from 10 March 2013, with $10\,\text{min} \times 7.5\,\text{m}$ resolution, excluding anywhere with more than 0.2 absolute uncertainty in depolarization parameter units. Top panel: $d_2$ values. Centre panel: The uncertainties associated with the $d_2$ values in units of depolarization parameter. Lower panel: The relative uncertainties are in units of percent. The calibration profile is based on $20\,\text{min} \times 7.5\,\text{m}$ resolution calculations.





The deficiencies of $d_1$ and advantages to using $d_2$ are clearly seen. There are large differences in data

10  coverage at this higher resolution, and there are features visible in $d_2$ which are not visible in $d_1$: $d_1$ can

barely discern that there is a cloud at all at 13:30:00 UTC, while $d_2$ still clearly gives the cloud's shape.

## 6  Importance of calibration selection region

The calibration profile $Y(z)$ can only be calculated based on valid values of $d_1$, which themselves are

only possible to calculate in regions where the lidar assumptions of single scattering and low extinction

15  are valid. In some meteorological cases, these assumptions are not appropriate. The following example

illustrates the importance of selecting an appropriate region in time and altitude for the calibration. A co-

adding resolution of $20\,\mathrm{min} \times 7.5\,\mathrm{m}$ was selected for this comparison. This provides sufficient photons in

the perpendicular channel to make calculations.

On 14 March 2013, shown in Fig. 11, there are clouds above $6000\,\mathrm{m}$ which descend (or, more likely,

clouds at lower altitude move over the lidar) gradually over the day. By 8 hours later, the clouds have

5  become thick enough to extinguish light backscattered above the cloud, and perhaps even from the upper

regions of the cloud. The cloud descends to the ground by 12:00:00 UTC and remains there for many

hours. Values of $d_1$ will not be valid in the thick cloud, and are not calculable at all above it.

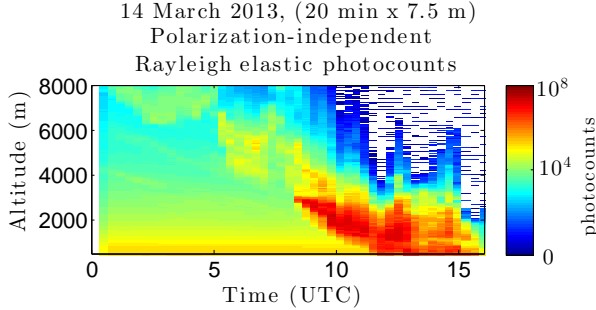

**Figure 11.** Polarization-independent Rayleigh elastic photocounts from 14 March 2013. Note the logarithmic colourbar used for this plot. These photocounts have been dead-time corrected, coadded, and background-subtracted.



The $d_1$ depolarization parameters are presented with uncertainty and relative uncertainty in Fig. 12. Individual $Y(z)$ calculations are made next for each time-altitude measurement point for the entire time and altitude range for this day. These are plotted in Fig. 13. The analysis of determining a representative calibration profile for this day, and using it to calculate $d_2$, was performed twice: Once including all the data (Box A in Fig. 13; Panel a in Fig. 14; Fig. 15), and again taking into account only regions without thick clouds (Box B in Fig. 13; Panel b in Fig. 14; Fig. 16). We can see at once that using the whole region (Box A) will not be appropriate: the value of $Y(z)$ at particular altitudes, such as at 1000 m, is not constant throughout the time series.

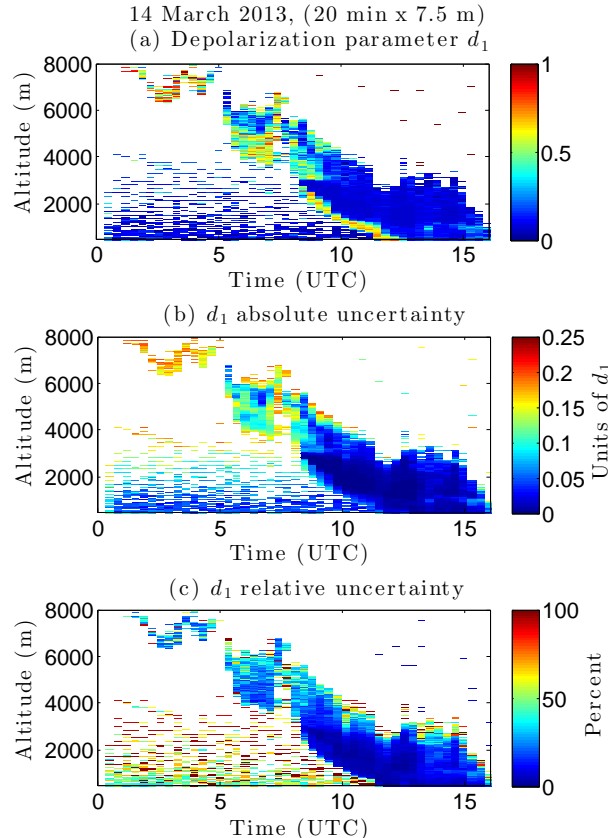

**Figure 12.** Upper panel: The $d_1$ depolarization values from 14 March 2013, at 20 min × 7.5 m resolution. Centre panel: The uncertainties associated with the $d_1$ values are all in units of depolarization parameter. Lower panel: The relative uncertainties are in units of percent. Values indicated for the interior of the optically thick cloud are not valid, although they have been calculated.





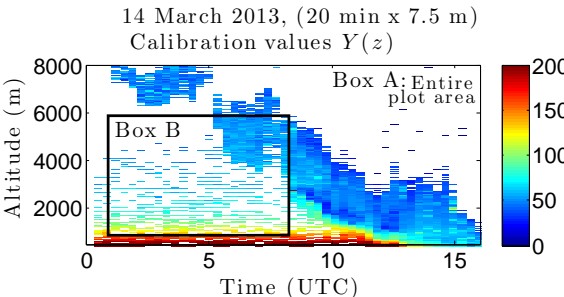

**Figure 13.** Context plot of all individual $Y(z)$ values for 14 March 2013. Box A indicates the region included in the nightly profile which includes all measurements. The profiles from this selection are given in Panel (a) of Fig. 14. Box B indicates the region included in the nightly profile which excludes any regions with thick clouds. This clear-sky region has profiles given in Panel (b) of Fig. 14.

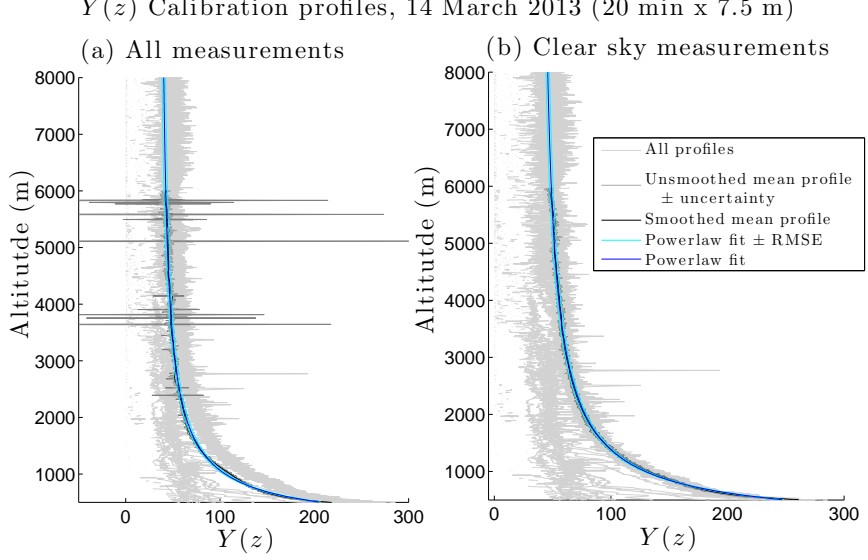

**Figure 14.** Profiles of fits to $Y(z)$ calibration values for 14 March 2013. Panel (a) uses profiles from all measurements from the measurement period, which corresponds to those in Box A of Fig. 13. Panel (b) uses only profiles which include clear sky, which are those indicated in Box B of Fig. 13. All regions with thick clouds have been excluded.





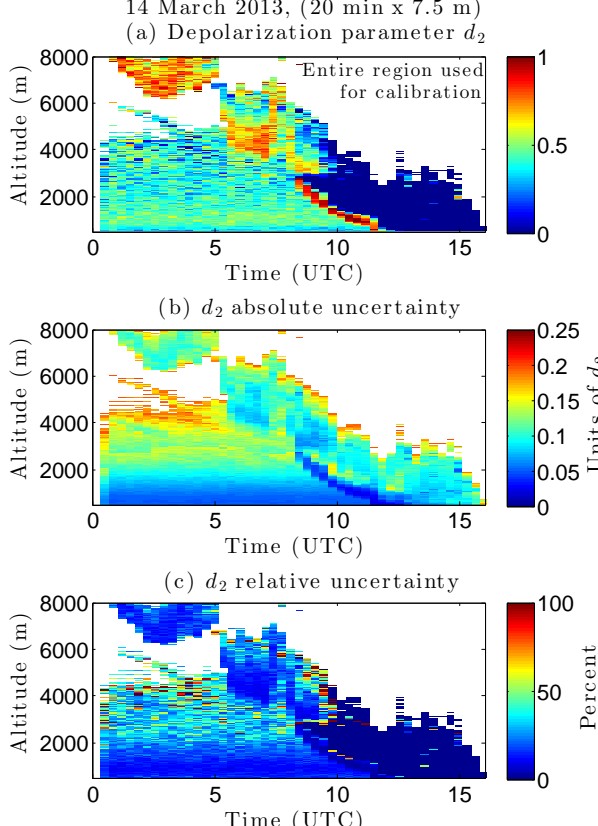

**Figure 15.** Upper panel: The $d_2$ depolarization values from 14 March 2013, using all data available for that day to influence the calibration profile. Centre panel: The uncertainties associated with the $d_2$ values are all in units of depolarization parameter. Lower panel: The relative uncertainties are in units of percent.

Figure 15 illustrates the perils of blindly choosing a calibration region of sky. The "default" region, Box A in Fig. 13 which encompasses the entire data region, does a poor job of producing $d_2$ values which mimic those given by $d_1$ in Fig. 12. Furthermore, the resulting $d_2$ values show high depolarization in regions of clear air during the first half of the day, which is incorrect. An examination of the photocount plot (Fig. 11) reveals that the current example has a bad choice of calibration region.

Consider next the plots of Fig. 16, in which a more careful calibration region, Box B, was selected. The entire region with the thick cloud has been excluded from the calculation of the calibration profile. Then





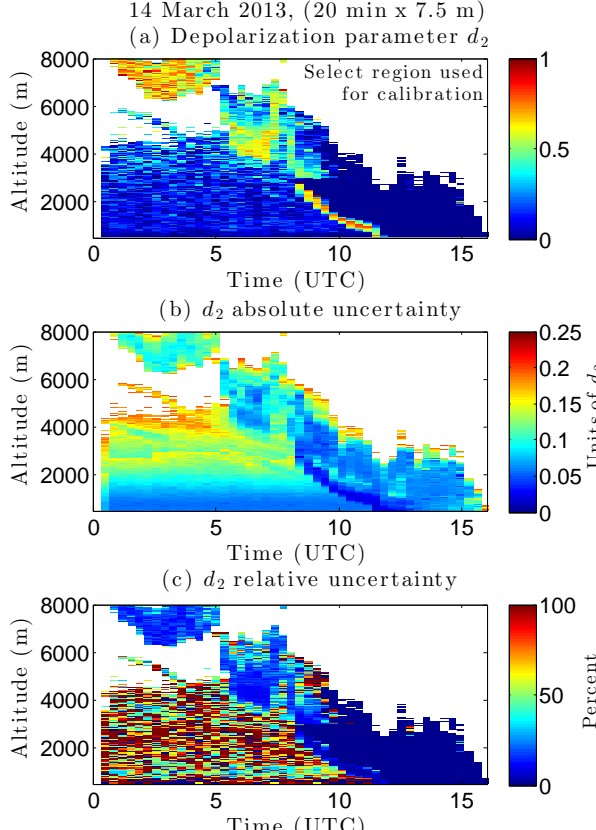

**Figure 16.** Upper panel: The $d_2$ depolarization values from 14 March 2013, using only clear sky and regular cloud data (thick clouds excluded) to influence the calibration profile. Centre panel: The uncertainties associated with the $d_2$ values are all in units of depolarization parameter. Lower panel: The relative uncertainties are in units of percent.

this conservative profile has been applied to the entire data space to produce the $d_2$ plot. An examination of the difference between the $d$ values shows that in almost all cases for the beginning of the measurement, the values of $d_1$ and $d_2$ are the same to within $\pm 0.1$ in depolarization parameter units. The only location which does not match well (though differences are no larger than those when using Box A) is within the thick cloud where the original $d_1$ measurements are invalid to start with.

Most of the measurement is good; within the upper regions of the thick cloud, it is not. Regions of CRL plots which are least trustworthy in terms of atmospheric interpretation are those which lie above strongly scattering layers. Although there is well-understood precision regarding the number of photons returned from these higher regions to the lidar (and therefore the calculated uncertainty is quite low), their





interpretation is less clear as multiple scattering cannot be ruled out. The $d_2$ calibration procedure is not

handled autonomously at this time, as the calibration region must be selected by hand. Some care must be

taken to understand which data are trustworthy, which are less so, and the geophysical reasons for this.

## 7   Discussion

The depolarization parameter obtained from our new method, $d_2$, is able to retain more useful measure-

ments with lower uncertainties than are possible with the depolarization parameter $d_1$ from the traditional

method for CRL. It can give depolarization parameter information to much higher altitudes (frequently

twice as high), and for the same resolution loses fewer data points to counting noise. Alternatively, $d_2$ may

be used at much higher resolutions, thus resolving fine scale structure to which $d_1$ is not sensitive.

Given that depolarization parameter measurements are used in a wide variety of applications in the

context of global atmospheric science, improvements to the quality of $d$ measurements help in several

ways. First, for comparison measurements with other instruments having high temporal and/or spatial

resolution, an increase in lidar $d$ resolution will be important. Comparisons with Eureka's starphotometer

measurements and other lidar measurements of fine aerosol layers, or very thin clouds, will be possible if

the lidar depolarization parameter measurements are available for such fine structures as they are with $d_2$,

but not with the $d_1$ data product. Further, many microphysical processes (e.g. evaporation, sublimation,

deposition, ice crystal growth, etc.) happen in thin layers or small regions within a cloud; it is desirable

that depolarization parameter measurements be sensitive at these spatial scales, which are on the order

of metres. Low uncertainty is vital if one is to examine small differences in the depolarization parameter

within specific clouds. The increased altitude range of the $d_2$ measurements has different advantages.

There are instruments at Eureka which measure whole-column quantities (having no altitude resolution).

The $d_2$ measurements to higher altitudes, capturing more of the relevant clouds and aerosols in its data

(including those missed by $d_1$, but which are certainly captured by the whole-column instruments), will

allow a more reasonable comparison with these range-integrated data products. Finally, once sufficient

depolarization measurements have been made, survey-type investigations may be done to examine the

relative frequency and coverage of various types of clouds; this can only be done well if the lidar can see

the clouds. This is bound to be a more thorough survey when done using the $d_2$ product than it is using

the $d_1$ product which misses data from many regions of the atmosphere.





The first example showing the advantages of using $d_2$ rather than $d_1$ is 10 March 2013, which shows one cloud near the start of the day extending from 3000 m to 5000 m altitude (see Figs. 2, 7, 9, and 10). There are several smaller clouds between 5000 m and 6000 m a few hours later. The $d_2$ measurements are required to identify fine scale cloud structure and allow depolarization parameter to be determined at low altitudes.

The second measurement example is 14 March 2013 (see Figs. 12 and 16). This date was selected for its different meteorology as compared to 10 March 2013, as there are optically thick clouds with large vertical extent on 14 March 2013. 14 March 2013 begins with one cloud above 6500 m lasting until 05:00 UTC. $d_1$ can nearly discern the cloud bottom height, but it cannot give information further up in the cloud. $d_2$ has lower uncertainty, and clearly shows this to be an ice cloud extending to at least 8000 m. Beneath this cloud, there are some layers of higher backscatter and moderate depolarization descending in an arc from 6000 m at 01:00 UTC to 4000 m at 04:00 UTC. $d_2$ is sensitive to these layers, while $d_1$ is not. Finally, beginning at 05:00 UTC, there is an optically thick cloud between 4000 m and 6000 m (and potentially higher) which reaches the ground by 11:00 UTC. This optically thick feature remains until the end of the measurement period.

The optically thick cloud on 14 March 2013 is an example of meteorology which requires more care when determining $Y(z)$ so that $d_2$ can be calculated correctly. Therefore, this date was used to demonstrate the different outcomes for $d_2$ when using the entire 15 h dataset in the calculation of the calibration profile $Y(z)$ (ineffective; provides incorrect $d_2$ values for the whole measurement period) versus using a more appropriate subsection of the dataset to calculate $Y(z)$ and then applying the profile to the entire dataset (effective; provides correct $d_2$ values for all measurement periods for which it is appropriate to calculate them). Neither the $d_1$ nor the $d_2$ values are appropriate to use for atmospheric interpretations anywhere that multiple scattering is expected to occur. For example, at 2000 m at 14:00 UTC, values for $d_1$ and $d_2$ can be calculated, but they are likely to be the result of multiple scattering within the thick cloud, and should be discounted from calibrations and from depolarization interpretations.

It is recommended that users of CRL depolarization measurements make use of the $d_2$ depolarization parameter measurements. These are available at higher resolution and lower uncertainty than traditionally-calculated $d_1$ depolarization parameter measurements. For CRL, the highest quality depolarization measurements are the depolarization parameter values calculated from the parallel and polarization-independent Rayleigh elastic channels, calibrated nightly using contributions from the perpendicular channel.





If depolarization ratio measurements are desired instead of depolarization parameter, the $d_2$ measurements may be easily converted into expressions for that quantity, according to standard methods (see e.g. McCullough et al. (2017), after Gimmestad (2008)).

## 8 Conclusions

In McCullough et al. (2017) the addition of a linear depolarization system to the CANDAC Rayleigh-Mie-Raman lidar (CRL) at Eureka, Nunavut in the Canadian High Arctic was discussed. Calibrated measurements of the depolarization ratio were shown produced according to the methods which are common in the depolarization lidar community. Calculations of the related depolarization parameter were also made. These methods are based on a ratio of the parallel and perpendicular depolarization channel measurements.

In an extension of McCullough et al. (2017), we have shown here that matrix calculations open up a new possibility for CRL depolarization measurements: the use of a third, non-polarized Rayleigh elastic lidar channel. In this work, we developed equations for the calculation of the depolarization parameter using combinations of the three available channels (two polarization-dependent, one polarization-independent), and these were expressed in terms of the fewest calibration constants possible.

For the most promising depolarization calculation option, full worked examples were presented using CRL measurements from 2013. In these examples, the parallel and perpendicular channel measurements were used at low resolution to calculate a calibration profile for the night. Then the parallel and polarization-independent elastic channels were used at high spatial-temporal resolution, along with the nightly calibration profile, to produce estimates of the depolarization parameter.

The advantages of the new three-channel calculation technique are several relative to the traditional method: better coverage in time and space, and higher spatial and temporal resolution of derived the depolarization parameter data products, due to higher signal-to-noise ratios.

CRL depolarization measurements exist for 2013, 2014, 2016 and 2017 with at least one month (in some cases, more than two) of approximately $24\,\mathrm{h/d}$ coverage in the polar sunrise season of each year, taken with the same settings as the measurements presented in this paper. Now that measurements have been optimized and so have the calibrations, routine calculations of depolarization ratio and depolarization parameter plots for these years, with uncertainties, will be produced.





The low perpendicular signals and very large $k$ value of the $d_1$ method for the CRL were the reason for our development of the $d_2$ method. The advantages of the $d_2$ method apply to other lidar systems as well, including those with $k \approx 1$, provided their polarization-independent channel has signal rates larger than those in either of the other two channels. An extension to this method is simple to apply in the case that $k << 1$ for a particular lidar: in that case, the algebra would be carried out to eliminate the parallel

channel measurements from the high-resolution calculations. The relative signal rates in a particular lidar's measurement channels will indicate whether the advantages of the $d_2$ method are significant enough to warrant its use at that laboratory. Likewise, practical considerations will determine whether the use of the two-channel $d_2$ method, calibrated nightly without the use of $d_1$, will determine whether that procedure is useful to any other lidar, or whether as for CRL, the three-channel $d_2$ procedure is of more benefit.

## 9   Data availability

Data used in this paper available upon request from corresponding author (e.mccullough@dal.ca).

**Appendix A:  Demonstration that CRL's Rayleigh Elastic Channel is polarization independent**

Measurements in all three $532\,\mathrm{nm}$ channels were made on 5 March 2014 during a calibration test in which a cube polarizer was mounted at the entrance to the polychromator, just downstream of the focus stage in the lidar (Fig. 1 of McCullough et al. (2017)). This polarizer was rotated, and lamp light was shone first through a depolarizing glassine sheet, and then through the polarization-generating cube polarizer. The cube polarizer was rotated to a variety of angles $\theta$, and the signals in each channel were measured as a function of angle. Any channel whose response is not sensitive to the rotation angle of the polarization-

generating cube polarizer is considered to be a "polarization independent channel".

The measurements from this calibration are presented in Fig. 17. The measurements shown for the Parallel and Perpendicular channels have already been presented in Fig. 2 of McCullough et al. (2017), and further details of the calibration procedure are available in that paper. The polarization-independent Rayleigh elastic results are added in the present figure.





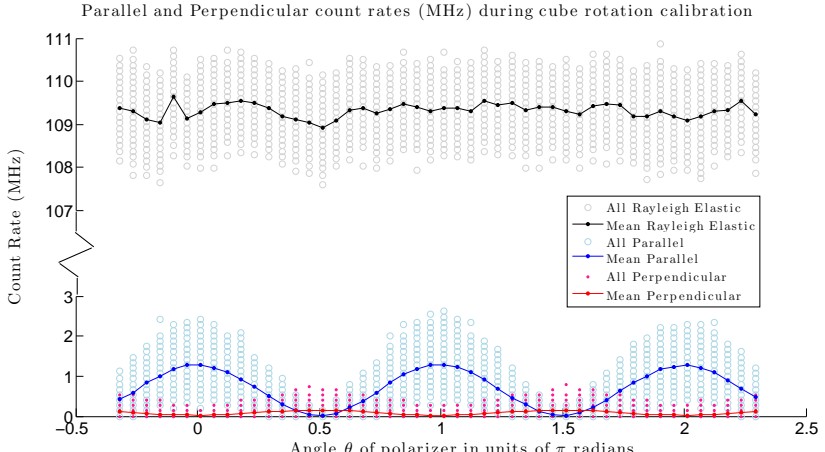

**Figure 17.** Polarized calibration measurements as a function of incident light polarization angle for all channels. Note the broken axis; the polarization-independent Rayleigh elastic measurements (grey circles and black line) are an order of magnitude larger than the Parallel (light blue circles and blue line) and Perpendicular (pink points and red line) measurements.

Equation (A1) gives the signal in the polarization-independent Rayleigh elastic channel as a function of polarizer rotation angle $\theta$:

$$S_{R\theta} = G_{cube} G_{PMTR} G_{gl} \frac{I_{lamp}}{2} \left( T_{00} + T_{01} \cos 2\theta + T_{02} \sin 2\theta \right). \tag{A1}$$

in which: $S_{R\theta}$ is the signal rate measured in the polarization-independent Rayleigh elastic channel as a function of the rotation angle $\theta$ of the polarization-generating cube polarizer, $G_{cube}$ is the attenuation of the polarization-generating cube polarizer, $G_{PMTR}$ is the gain (or attenuation) of the photomultiplier tube, $G_{glassine}$ is the attenuation of the depolarizing glassine sheet, $I_{lamp}$ is the lamp intensity, $T_{xx}$ are individual elements of the 4×4 Mueller matrix **T** which describes the combined optical effect of all optics between the polarization-generating cube polarizer and the polarization-independent Rayleigh elastic PMT.

**A1   First result: Symmetry, $T_{02} = 0$**

$T_{02}$ is zero if there is symmetry in the curve of the signal with angle (i.e. the values at $\theta = \frac{\pi}{4}$ equal those at $\theta = \frac{3\pi}{4}$ for the polarization-independent Rayleigh elastic channel). Examining the measurements, it is



evident that this is true for the polarization-independent Rayleigh elastic channel; the measurements at
multiples of $\theta = \frac{\pi}{4}$ equal those at multiples of $\theta = \frac{3\pi}{4}$.

$T_{02}$ does not appear in any equations for the depolarization parameter $d$ shown in the paper, but its
determination here allows the calculation of $T_{01}$, which does appear in the expression for $d$.

### A2  Second result: Constant signal with angle, $T_{01} = 0$

All angle-dependence information in the polarization-independent Rayleigh elastic channel's signal equa-
tion is contained within the term including calibration constant $T_{01}$. If it is the case that the measurements
do not vary with polarizer angle, it may be inferred that $T_{01} = 0$.

An examination of Fig. 17 demonstrates that this is the case. The mean at each angle is not statistically
significantly different from the mean at any other angle. Therefore, the CRL (for optics downstream of
the focus stage, at least) has the calibration coefficient $T_{01} = 0$. There is no polarization dependence in
this channel. The resulting signal equation for this channel is:

$$S_{R\theta} = G_{cube}G_{PMTR}G_{gl}\frac{I_{lamp}}{2}\left(T_{00}\right). \tag{A2}$$

As the individual gains of the PMT, the cube polarizer, the glassine, and the intensity of the lamp remain
unknown throughout the test, it is not possible to determine $T_{00}$ by rearranging this equation and solving
for it.

Results indicating that both $T_{01}$ and $T_{02}$ are zero are encouraging for the CRL. Considering that this
channel is intended to be polarization-independent, these results are what one would expect. If its mea-
surements indicated a polarization preference, then the CRL's Rayleigh Backscatter Coefficient data prod-
ucts, and all others using this channel, would need to be re-evaluated. Fortunately, the channel performs
as intended.

*Author contributions.*  E. M. McCullough: Installation and calibration of depolarization hardware. Data analysis and development of method.
Writing of analysis MATLAB code. Manuscript preparation. This work formed part of McCullough's doctoral thesis. R. J. Sica: Supervision
5  of doctoral thesis. Contribution to manuscript preparation. J. R. Drummond: Principal Investigator of PEARL laboratory. Contribution to
manuscript preparation. T. J. Duck: Principal Investigator of CRL lidar at the time of this work. Development of the CRL laboratory.
G. J. Nott: Original design of depolarization channel for CRL. Instruction in installation and initial data processing in Python. Extensive
discussions regarding calibration and analysis. Contribution to manuscript preparation. C. Perro: Lidar operations and remote laboratory
assistance.



10  *Competing interests.* The authors declare that they have no conflict of interest.

*Acknowledgements.* PEARL has been supported by a large number of agencies whose support is gratefully acknowledged: The Canadian Foundation for Innovation; the Ontario Innovation Trust; the (Ontario) Ministry of Research and Innovation; the Nova Scotia Research and Innovation Trust; the Natural Sciences and Engineering Research Council; the Canadian Foundation for Climate and Atmospheric Science; Environment and Climate Change Canada; Polar Continental Shelf Project; the Department of Indigenous and Northern Affairs Canada; and the Canadian Space Agency. This work was carried out during the Canadian Arctic ACE/OSIRIS Validation Campaigns of 2010, 2011, 2012 and 2013, which are funded by: The Canadian Space Agency, Environment and Climate Change Canada, the Natural Sciences and Engineering Research Council of Canada, and the Northern Scientific Training Program. This particular project has also been supported by NSERC Discovery Grants and Northern Supplement Grants held by J. R. Drummond, R. J. Sica, and K. A. Walker, and the NSERC CREATE

5  Training Program in Arctic Atmospheric Science (PI: K. Strong). In addition, the authors thank the following groups and individuals for their support during field campaigns at Eureka: PEARL site manager Pierre Fogal; Canadian Arctic ACE/OSIRIS Validation Campaign project lead K. A. Walker, CRL operators Colin P. Thackray, Jason Hopper, Shayamila Mahagammulla Gamage, and Jon Doyle, Canadian Network for the Detection of Atmospheric Change (CANDAC) operators: Mike Maurice, Peter McGovern, Alexei Khmel, Paul Leowen, Ashley Harret, Keith MacQuarrie, Oleg Mikhailov, and Matt Okraszewski; and the Eureka Weather Station staff.





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





**Table 1.** Fitting coefficients (a, b, and c; fitting a second-order power law of the form $y = ax^b + c$) and goodness of fit ($R^2$ and root mean square error (RMSE)) for various dates, time resolutions (time res.) and altitude resolutions (alt. res.) in March 2013 used for determining calibration function $Y(z)$. Days marked with "*" used only a portion of the data available for that day: clear sky or thin clouds only.

| Test | Date | time res. (min) | alt. res. (m) | a (bounds) | b (bounds) | c (bounds) | $R^2$ | RMSE |
|------|------|-----------------|---------------|------------|------------|------------|-------|------|
| i | 10 Mar | 20 | 7.5 | 115200 (108200,122300) | -1.026 (-1.036,-1.017) | 31.81 (31.29,32.34) | 0.998 | 1.523 |
| ii | 10 Mar | 10 | 35.5 | 48260 (29866,67870) | -0.8896 (-0.9555,-0.8238) | 27.36 (22.72,32) | 0.982 | 4.750 |
| iii | *10 Mar | 10 | 7.5 | 92540 (86810,98280) | -0.9876 (-0.9975,-0.9777) | 26.04 (25.54,26.62) | 0.998 | 1.594 |
| iv | *11 Mar | 20 | 7.5 | 164900 (154200,175500) | -1.085 (-1.095,-1.075) | 35.04 (34.55,35.53) | 0.998 | 1.542 |
| v | 11 Mar | 2 | 15.0 | 310200 (230500,390000) | -1.172 (-1.213,-1.131) | 26.44 (24.62,28.26) | 0.985 | 4.753 |
| vi | *14 Mar | 20 | 7.5 | 567700 (514400,621000) | -1.304 (-1.318,-1.289) | 35.71 (35.29,36.12) | 0.996 | 1.860 |
| vii | *14 Mar | 20 | 7.5 | 407900 (383400,432400) | -1.217 (-1.227,-1.208) | 38.6 (38.21,38.98) | 0.998 | 1.504 |