# Peer review of "Three-channel single-wavelength lidar depolarization calibration"

_Atmospheric Measurement Techniques, 2017_

## Referee Comment (RC1) · Anonymous Referee #1 · 8 Nov 2017

The discussion paper "Three-channel single-wavelength lidar depolarization calibration" presents an interesting approach to retrieve depolarization parameters without the use of the cross-depolarization channel. The technique presented could be of interest to some lidar systems, and is clearly presented. I suggest the publication of the manuscript after some minor corrections.

General comments

The manuscript would benefit from a brief description / schematic of the lidar receiver setup. This will help the readers understand the discussion, without referring to previously published papers of the authors.

The authors use the depolarization parameter d, which is fine. Still, much of the liter-

ature refers to delta parameter of the value, and this makes it hard to evaluate if the d values presented in the study. The manuscript will benefit from few examples linking d to delta (e.g. for d=0.1, d=0.6 etc.).

The authors should explain more clearly the disadvantages of not using the cross-polarization channel for depolarization calculation. Why bother with this channel if all information can be derived equally well with just total and parallel channel?

The authors assume that there is no cross-talk between parallel and perpendicular channel. (M11 = M00, M01 = M10). Small cross-talk could be easily caused by mis-alignment of the polarization separation element with respect to the laser polarization plane. Consequently, the authors should provide some calculation / graph describing the effect of possible crosstalk to the accuracy of the retrieved depolarization parameters for different values. This will help the readers assess the required accuracy for implementing this setup in their system.

The comparison scheme described in Section 5.5 seems problematic. The authors exclude points with uncertainty greater than 0.2. Is this limit applied to either d1 or d2 or both? In any case this could hide regions where one of the two methods provide better results. Please consider adding in Figure 8 a scatter plot including all points (without the 0.2) threshold. The two plots combined will give a better idea of the performance of the proposed method.

The number of figures could be grouped by grouping some of them (e.g. fig 3 and 4, 15 and 16, . . .).

Minor comments Page 5, line 4: gone undergone.

Page 9, line 14: give examples of what other factor could influence the overlap function

Page 16, line 7: scan-by-scan. Do you mean profile-by-profile?

Page 17, line16: do you have any theoretical reason why a power low would fit the calibration data? Do you expect such law to work also for other systems?

---

## Referee Comment (RC2) · Anonymous Referee #2 · 9 Nov 2017

Reviewer Comments

Title: Three-channel single-wavelength lidar depolarization calibration
Author(s): Emily M. McCullough et al.
MS No.: amt-2017-328

Summary

The authors describe a depolarization technique that is specific to a particular lidar, which has been equipped with a linear depolarization measurement capability (parallel and perpendicular receiver channels) by means of a Licel Polarotator. Unfortunately, the lidar system optics attenuate the perpendicular channel by a factor of 21, and so extensive signal averaging is required to get a useful SNR in that channel, which causes such low temporal and spatial resolution in the depolarization data that it is not useful for the cloud studies that the authors would like to enable.

However, the lidar also has a high-SNR polarization-insensitive receiver channel, and so the authors show how to use that channel with the parallel channel to derive the depolarization parameter. But they desire to measure depolarization in the crossover region, and the overlap functions of the two receiver channels involved are not the same, and differences are sensitive to laser beam alignment and even laboratory temperature. So, using this method would require time-consuming calibration measurements every night, which is undesirable.

The authors present an optimum approach, using low-resolution results from the Polarotator channels, which are stable, to calibrate the high-resolution results. They illustrate the benefits of their approach with the color time-height diagrams.

General Comments

The manuscript is very well written. The arguments and the formalism in the paper appear to be correct. The manuscript is very thorough, and the technique is well illustrated with the color time-height diagrams. This paper is a follow-on to a previous paper, and it is properly set in that context.

Specific Comments

Here are some corrections and suggestions:

p. 2 line 16 reads "The maximum signal in the parallel channel would be much greater than the maximum signal in the perpendicular channel, even without the partial-polarizer effects of the CRL's receiver optics." Well, usually. Not always, though. Maybe add the qualifier "usually"?

p. 3 line 3 might read better as "… light of all polarizations."

p. 3 line 27 should read "… and it can be…"

p. 8 line 4 should read "…to develop an expression…"

p. 9 line 1 should read "…which cannot be…"

p. 9 line 17 should read "…known, from…" (or else delete "which found" in the next line).

p.10 line 4 should read "None of the parameters…"

p. 11 line 25 should read "It is possible to circumvent $d_2$'s calibration disadvantages…"

p. 12 line 8 might read better as "…is to combine these two methods…"

p. 17 line 14 should read "A number of options…" or "Several options…"

p. 17 line 15 "powerlaw" should be two words.

p. 24 line 7 "…no longer able to be discerned." would read better as "…no longer discernable."

p. 26 line 13 "…which themselves are only possible to calculate…" would read better as "…which can be calculated only in…"

p. 32 line 12 should read "…allow the depolarization…"

p. 33 line 15 "…depolarization ratio were shown produced…" should read "…depolarization ratio were produced…"

p.33 line 30 "…derived the depolarization…" should read "…derived depolarization…"

---

## Author Comment (AC1) · 18 Dec 2017

**Author response to AMT review - first stage**
**(Response to reviewer comments)**
**Atmos. Meas. Tech. Discuss., doi:10.5194/amt-2017-328-RC1, 2017**

Three-channel single-wavelength lidar depolarization calibration. Emily M. McCullough1,2,*, R. J. Sica1, J. R. Drummond2, Graeme Nott2,a, Christopher Perro2, and T. J. Duck2

**Referee comments:**

**Referee #1:**

R1 General comment 1: The manuscript would benefit from a brief description / schematic of the lidar receiver setup. This will help the readers understand the discussion, without referring to previously published papers of the authors.

Response: We will add in a schematic, with a description of the system in the caption.

Action: After the sentence on page 2 which reads "Linear 532 nm depolarization capabilities were added to the lidar in 2010 with the addition of a beamsplitter, a Licel Polarotor rotating polarizer, and a photomultiplier tube detector.", insert the new sentence: "Figure 1 is a schematic of the lidar receiver in this configuration."

The new figure will be very similar to that in McCullough 2017 and Nott 2012. The caption should read:

"Diagram of CRL's receiver system.The Visible long wave pass filter picks off approximately 97% of the received 532 nm light and directs it toward the Visible Rayleigh Elastic (polarization-independent) channel, and transmits the remainder toward the Depolarization and Visible nitrogen channels downstream. The pellicle beamsplitter reflects 50% of this residual 532 nm light toward the Polarotor and into the Depolarization channels' detector (Parallel and Perpendicular polarization-sensitive channels, measured using the same PMT on alternate laser shots). The depolarization hardware which was added in 2010 consists of the pellicle beam splitter, the Polarotor, and the interference filter, focusing lens, and PMT, to the right of the Polarotor in this figure (McCullough 2017). No other receiver optics were modified so as to reduce the impact of this depolarization upgrade on the pre-existing lidar channels. A consequence of this choice is that the count rates are much higher in the Visible Rayleigh Elastic polarization-insensitive channel than they are in either the parallel or the perpendicular channel. This figure is based on Fig. 2 of Nott et al. (2012)".

[Figure]

R1 General comment 2: The authors use the depolarization parameter d, which is fine. Still, much of the literature refers to delta parameter of the value, and this makes it hard to evaluate if the d values presented in the study. The manuscript will benefit from few examples linking d to delta (e.g. for d=0.1, d=0.6 etc.).

Response: We will add in a few such examples.

Action: On page 4, just after line 23 (at the very end of Section 1, which ends "The development shown here is

easily adaptable to any similar lidar, and to any lidar with a single unpolarized, and single polarized channel."), add in a new paragraph:

The new paragraph will read: "An alternative representation of the depolarization is the depolarization ratio $delta$ = d/(2-d). The development in this paper is in terms of $d$ but values of $delta$ will be supplied where relevant".

On page 14, line 14, just after the sentence "Both expressions for uncertainty are useful in interpreting the depolarization values.", add a new sentence which reads "This measured depolarization parameter value of $d1$ = 0.6 corresponds approximately to a depolarization ratio value of $\delta$ = 0.43.".

On page 22, line 2, at the end of the sentence "Some regions do have d2 relative uncertainty approaching 100 %, but these are regions with depolarization parameter values < 0.1.", add text such that the sentence reads "Some regions do have d2 relative uncertainty approaching 100 %, but these are regions with depolarization parameter values < 0.1 (which corresponds to a depolarization ratio value of $\delta$ < 0.05)".

R1 General comment 3: The authors should explain more clearly the disadvantages of not using the cross-polarization channel for depolarization calculation. Why bother with this channel if all information can be derived equally well with just total and parallel channel?

Response: Depolarization d2 can be derived with just total and parallel - but not nearly as well, and not practically, at CRL.

A calculation which totally excludes the perpendicular channel would require a time-consuming and personnel-intensive calibration procedure each night to account for the differential overlap between the parallel and polarization-independent channels. This calibration is hours long, and requires a 1-metre calibration optic to be installed at the beginning and uninstalled later. This is not feasible at CRL, and is not expected to be desirable at other laboratories. The calibration measurements cannot be made simultaneously with "regular measurements" in any lidar channels, so valuable atmospheric measurements in all channels are lost during the calibration time. Page 11 lines 5 through 25 already includes some of these details.

Times that the two-channel method could be more useful than the three-channel method: If, for some reason, perpendicular measurements are totally impossible at a laboratory, then the two-channel parallel/polarization-independent procedure can of course be carried out without the perpendicular channel. Additionally, if a different lidar were to have full overlap starting from very low altitudes, perhaps the two-channel procedure without the perpendicular channel would be adequate, if measurements below the altitude of full overlap could be discarded with no ill effects.

Action: Add a sentence to Page 3 line 11 in the introduction, to make the point early in the manuscript that CRL finds the three-channel method to be most useful. End the following paragraph:
"The main advantages of the methods presented here are as follows: 1. We can determine d excluding the low-SNR polarization-dependent channel altogether. 2.We have the flexibility to include simultaneous information from the low-SNR polarization-dependent channel (the perpendicular channel for CRL) at low resolution to calibrate and improve the high time-altitude resolution d from the high-SNR polarization-dependent channel (the parallel channel for CRL) and the high-SNR polarization-independent channel."
with the new text:
"At CRL, the calibrations required to use the former (the two-channel method) are not particularly practical during routine operations, while use of the latter (the three channel method) is practical and brings benefits of higher temporal and/or spatial resolution to our measurements of depolarization".

R1 General comment 4: The authors assume that there is no cross-talk between parallel and perpendicular channel. (M11 = M00, M01 = M10). Small cross-talk could be easily caused by misalignment of the polarization separation element with respect to the laser polarization plane. Consequently, the authors should provide some calculation / graph describing the effect of possible crosstalk to the accuracy of the retrieved depolarization parameters for different values. This will help the readers assess the required accuracy for implementing this setup in their system.

Response: We realize the wording as currently have it implies that (M11 = M00, M01 = M10) is an assumption. In fact, we have done calibration measurements to verify these values. In McCullough2017a, we have described the Polarotor Start Delay Calibration which aligns the analyzing polarizer with the laser polarization plane. The uncertainty in angular alignment remaining after this calibration is negligible compared to the size of other uncertainties in CRL's calculations of d. Further, we have tested that both the parallel and perpendicular signals go to zero at their respective minima, when polarized light is introduced to the detector at a variety of test angles. We will rewrite this sentence to make it clear that we have measured this sufficiently, and refer the reader to

McCullough2017a for details about this.

Action: On page 9, starting with line 15, we rewrite the paragraph to reflect the proper tests.

Original paragraph:
Five calibration factors are thus needed: X1, X2, X3, X4, and X5 (placeholders in response document; will retain actual variables in the text). Some information is already known: from polarized and unpolarized white light characterization tests in McCullough et al. (2017), which found M01/M00 = 0.91 +/-0.002 for CRL. Thus, each channel has a different gain, indicated by M01 ≠ M00. Further, M11 = M00 and M01 = M10, indicating an absence of cross-talk between the parallel and perpendicular channels; no parallel-polarized light gets into the perpendicular profiles, and vice versa. Detailed characterizations carried out with polarized light introduced to the receiver at a variety of angles show that if there is any sensitivity to polarization in the "polarization-independent" Rayleigh elastic channel, this effect is orders of magnitude smaller than the uncertainty in routine lidar measurements and does not affect analyses (Appendix A)."...

New paragraph:
Five calibration factors are thus needed: X1, X2, X3, X4, and X5 (placeholders in response document; will retain actual variables in the text). Some information is already known. Polarized and unpolarized white light characterization tests in McCullough et al. (2017) found that M01/M00 = 0.91 +/-0.002 for CRL. Thus, each channel has a different gain, indicated by M01 ≠ M00. These tests also ensured that the parallel channel is well-aligned with the polarization plane of the outgoing laser beam, and demonstrated that M11 = M00 and M01 = M10, indicating an absence of cross-talk between the parallel and perpendicular channels; no parallel-polarized light gets into the perpendicular profiles, and vice versa. An extension to one of these detailed characterization tests was carried out for the polarization-independent channel. Polarized light introduced to the receiver at a variety of angles show that if there is any sensitivity to polarization in the "polarization-independent" Rayleigh elastic channel, this effect is orders of magnitude smaller than the uncertainty in routine lidar measurements and does not affect analyses (Appendix A)."...

R1 General comment 5: The comparison scheme described in Section 5.5 seems problematic. The authors exclude points with uncertainty greater than 0.2. Is this limit applied to either d1 or d2 or both? In any case this could hide regions where one of the two methods provide better results. Please consider adding in Figure 8 a scatter plot including all points (without the 0.2) threshold. The two plots combined will give a better idea of the performance of the proposed method.

Response: Yes, the uncertainty limit was applied to both. If d1 and/or d2 have uncertainty > 0.2, we cut out the data point pair of (d1,d2). When making the identical type of scatter plot, but not cutting out anything based on uncertainty in final d1 and d2 values, the plot looks substantially the same, and the result we draw from this is the same also (namely, that d2 can reproduce d1 satisfactorily, to within the limits that we can measure d1). We will replace the plot with one including all data points. We'll provide some details about which method (d1 versus d2) provides the better results as well.

Action: Replace Figure 8 with a new plot (now numbered Figure 9, because of the addition of a schematic which is a new Figure 1), in which we do not remove any points based on d1 or d2 uncertainty:

[Figure]

On Page 22, line 18, we change the paragraph to remove the first step " 1. Removal of any data points for d1 and d2 with absolute uncertainty greater than 0.2;"

such that the paragraph will now read:

"To make the comparison, each plot had the following steps applied in sequence: 1. Smoothing the d1 and d2 measurements by 3 x 3 moving average filter, for a smoothed resolution of 60min x 22.5m; 2. Removal of any data points which are surrounded on three or four sides by an empty data point, done recursively twice such that isolated groups of two data points will also be eliminated; 3. Removal of any data points which do not exist in both of the plots."

On Page 23, line 2, we replace all the following sentences:

Using data from the original d1 and d2 plots (Fig. 2 and Fig. 7, after doing only steps 1 and 4: removing points with uncertainty > 0.2, and removing any points which do not exist in d1 and d2, but doing no smoothing nor removing of isolated data points), 5 14705 of 16024 valid data points match to within their uncertainties (91.8%). Using data which has been processed through all four steps, including smoothing and removing isolated data points (such as the data used to make Fig. 8, this improves to 12941 of 13036 points (99.3 %).

by the following text (figure numbers increase because of newly-added Fig. 1):

Of all data points in the original d1 and d2 plots for this time-altitude region (Fig. 3 and Fig. 8), none contain only a d1 measurement, 65% contain only a d2 measurement, and 35% contain both a d1 and a d2 measurement. All data points for which there are both a d1 and a d2 measurement are included in Fig. 9. Of these, the measurements are significantly different from each other in 0.1% of cases. For the remainder of the measurements in Fig. 9 (99.9% of cases), d1 = d2 to within their uncertainties. Of these statistically equal cases, d1 has the lower uncertainty for 15% of the points (almost exclusively from below 3000 m altitude, where count rates in the perpendicular channel are high), while d2 has the lower uncertainty for the other 20% of the points (some points below 3000 m, but most are above that altitude). If we omit the smoothing step in the Fig. 9 analysis, the number of significantly different d1 and d2 values increases to only 3%.

R1 General comment 6: The number of figures could be grouped by grouping some of them (e.g. fig 3 and 4, 15 and 16, . . .).

Response: The authors agree that keeping figures (old number 3 and 4, which now are new number 4 and 5) together would be beneficial. Placing these side by side was not possible based on the limitations for figure widths and minimum sizes for figure fonts given by AMT as we understand them, but the authors are happy if these are put closer together when the paper is typeset. Likewise, figures (old number 15 and 16, new number 16 and 17) would benefit from being directly comparable on a single page. Figures (old number 9 and 10, new number 10 and 11) could similarly be placed together.

Action: The final locations of the figures will need to be determined during typesetting, but the authors will endeavour to have the figures placed together in sensible groups if possible.

R1 Minor comment: Page 5, line 4: gone undergone.

Response: Done.

R1 Minor comment: Page 9, line 14: give examples of what other factor could influence the overlap function

Response: We will modify the paragraph to indicate that lab temperature, beam alignment on the laser table transmitter mirrors (translation on the mirrors), and beam alignment to the sky (which we adjust nightly) all influence the overlap function of the lidar. We have not investigated other contributions not related to geometric overlap since the effects must be calibrated away empirically in any case. Seeing as we do not have the means to determine our geometric overlap precisely, we cannot rule out other contributors to our altitude-dependent function, and thus choose to be general in our description.

Action: Modify the section which reads: "Overlap functions are in general difficult to determine for lidars. Here, the "overlap function" O(z) includes both geometric overlap (varies in altitude and time) as well as any other factors which vary in altitude (though they may be constant in time). The overlap function will be eliminated where possible, and available means will be used to determine it via calibration otherwise." such that it will now read "Overlap functions are in general difficult to determine for lidars. They tend to change with laboratory temperature (which we cannot control precisely), alignment of transmitter mirrors (adjusted at CRL approximately 2 to 3 times per year), and alignment of the laser beam to the sky (adjusted nightly at CRL). Here, the "overlap function" O(z) includes both geometric overlap (varies in altitude and time) as well as any other factors which vary in altitude (though they may be constant in time). The overlap function will be eliminated where possible, and available means will be used to determine it via calibration otherwise." ".

R1 Minor comment: Page 16, line 7: scan-by-scan. Do you mean profile-by-profile?

Response: Yes.

Action: Change the words "scan-by-scan" to "profile-by-profile" in the sentence "If there was good signal in the perpendicular channel, and thus good calibration measurements for each altitude, it would be possible to calibrate the lidar measurements scan-by-scan.", such that it will now read "If there was good signal in the perpendicular channel, and thus good calibration measurements for each altitude, it would be possible to calibrate the lidar measurements profile-by-profile."

R1 Minor comment: Page 17, line16: do you have any theoretical reason why a power low would fit the calibration data? Do you expect such law to work also for other systems?

Response: We do not have a physical reason to go with a 2nd order power law - but it seems to work better than the other standard functions that we tried for CRL, including exponentials. We suspect that most lidars will have a similar function: varying slowly at high altitudes, and increasing more rapidly at low altitudes. We know that the complete overlap is 1, so all lidars should eventually have their function approximately constant in altitude past some lower limit. Each group should look at their result to see whether this is optimal for their system. We tried a power law up to a certain altitude, and a constant after that, and for lidars which have a small overlap region this will probably work better than a 2nd order power law all the way up. But for CRL, the discontinuity this introduces was not tolerable, and we find the power law result to be more reasonable.

Action: Page 17 line 21, after the sentence "This fit has $R2 = 0:998$ and the root mean square error is $RMSE = 1:523$ (compared to the values of $Y(z) = 400$ at its largest point, and around 40 to 50 at its smallest).", insert a new sentence "For other lidars, a different function or combination of functions may be more reasonable to use for $Y(z)$.".

**Referee #2**

R2 Comment: p. 2 line 16 reads "The maximum signal in the parallel channel would be much greater than the maximum signal in the perpendicular channel, even without the partial-polarizer effects of the CRL's receiver optics." Well, usually. Not always, though. Maybe add the qualifier "usually"?

Response: Perhaps the issue is with the words "much greater" when we specifically mean "twice as large as". With a completely depolarized random return signal, the light will be split equally between the two orthogonally polarized detector channels, rendering the maximal perpendicular signal (half the total light; occurs when $d = 1$) only half as large as the maximal parallel signal (all the total light; occurs when $d = 0$). We also add in the word "usually" to account for other lidar scenarios.

Action:  Change the sentence  "The maximum signal in the parallel channel would be much greater than the maximum signal in the perpendicular channel, even without the partial-polarizer effects of the CRL's receiver optics." such that it now reads  "The maximum signal in the parallel channel would usually be be twice as large as the maximum signal in the perpendicular channel, even without the partial-polarizer effects of the CRL's receiver optics."

R2 Comment: p. 3 line 3 might read better as "… light of all polarizations."

Response: Done.

R2 Comment: p. 3 line 27 should read "… and it can be…"

Response: Done.

R2 Comment: p. 8 line 4 should read "…to develop an expression…"

Response: Done.

R2 Comment: p. 9 line 1 should read "…which cannot be…"

Response: Done.

R2 Comment: p. 9 line 17 should read "…known, from…" (or else delete "which found" in the next line).

Response: Done. We have also rewritten part of this paragraph to satisfy a General Comment from Reviewer 1, so please refer to "R1 General Comment 4", for p. 9, line 15.

R2 Comment: p.10 line 4 should read "None of the parameters…"

Response: Done.

R2 Comment: p. 11 line 25 should read "It is possible to circumvent d2's calibration disadvantages…"

Response: Done.

R2 Comment: p. 12 line 8 might read better as "…is to combine these two methods…"

Response: Done.

R2 Comment: p. 17 line 14 should read "A number of options…" or "Several options…"

Response: Done.

R2 Comment: p. 17 line 15 "powerlaw" should be two words.

Response: Done.

R2 Comment: p. 24 line 7 "…no longer able to be discerned." would read better as "…no longer discernable."

Response: Done.

R2 Comment: p. 26 line 13 "…which themselves are only possible to calculate…" would read better as "…which can be calculated only in…"

Response: Done.

R2 Comment: p. 32 line 12 should read "…allow the depolarization…"

Response: Done.

R2 Comment: p. 33 line 15 "…depolarization ratio were shown produced…" should read "…depolarization ratio were produced…"

Response: Done.

R2 Comment: p.33 line 30 "…derived the depolarization…" should read "…derived depolarization…"

Response: Done.

**Author additional comments:**

Following recent discussions with the AMT technical editor about affiliation numbering and addresses for McCullough 2017 (AMT, 10, 4253–4277, 2017), we have adjusted the same items for this paper to match the requirements.

Three-channel single-wavelength lidar depolarization calibration

Emily M. McCullough1,2,*, Robert J. Sica1, James R. Drummond2, Graeme Nott2,a, Christopher Perro2, and Thomas J. Duck2

[1] Department of Physics and Astronomy, The University of Western Ontario, 1151 Richmond St., London, ON, N6A 3K7, Canada

[2] Department of Physics and Atmospheric Science, Dalhousie University, 6310 Coburg Rd., PO Box 15000, Halifax, NS, B3H 4R2, Canada

[a] Present affiliation: Facility for Airborne Atmospheric Measurements, Building 146, Cranfield University, Cranfield, MK43 0AL, UK

*Correspondence to: Emily McCullough (emccull2@uwo.ca)

Author response to review - second stage: Atmos. Meas. Tech. Discuss., doi:10.5194/amt-2017-76-RC3, 2017